# Transferable Calibration with Lower Bias and Variance in Domain Adaptation

**Ximei Wang, Mingsheng Long,**[*] **Jianmin Wang, and Michael I. Jordan**[♯]
School of Software, KLiss, BNRist, Tsinghua University    [♯]University of California, Berkeley
wxm17@mails.tsinghua.edu.cn   {mingsheng,jimwang}@tsinghua.edu.cn
jordan@cs.berkeley.edu

## Abstract

Domain Adaptation (DA) enables transferring a learning machine from a labeled source domain to an unlabeled target one. While remarkable advances have been made, most of the existing DA methods focus on improving the target accuracy at inference. How to estimate the predictive uncertainty of DA models is vital for decision-making in safety-critical scenarios but remains the boundary to explore. In this paper, we delve into the open problem of *Calibration in DA*, which is extremely challenging due to the coexistence of domain shift and the lack of target labels. We first reveal the dilemma that DA models learn higher accuracy at the expense of well-calibrated probabilities. Driven by this finding, we propose Transferable Calibration (TransCal) to achieve more accurate calibration with lower bias and variance in a unified hyperparameter-free optimization framework. As a general post-hoc calibration method, TransCal can be easily applied to recalibrate existing DA methods. Its efficacy has been justified both theoretically and empirically.

## 1 Introduction

Deep neural networks (DNNs) achieve the state of the art predictive accuracy in machine learning tasks with the benefit of powerful ability to learn discriminative representations [35, 11, 57]. However, in real-world scenarios, it is hard (intolerably time-consuming and labor-expensive) to collect sufficient labeled data through manual labeling, causing DNNs to confront challenges when generalizing the pre-trained model to a different domain with unlabeled data. To tackle this challenge, researchers propose to transfer knowledge from a different but related domain by leveraging the readily-available labeled data, a.k.a. domain adaptation (DA) [44].

There are mainly two types of domain adaptation formulas: *covariate shift* [44, 37, 29, 13] and *label shift* [27, 2, 1], while we focus on the former in this paper since it appears more natural in recognition tasks and attracts more attention in the literature. Early domain adaptation methods bridge the source and target domains mainly by learning domain-invariant representations [37, 16] or instance importances [23, 15]. After the breakthrough in deep neural networks (DNNs) has been achieved, they are widely believed to be able to learn more transferable features [35, 11, 57, 61], since they disentangle explanatory factors of variations. Recent works in deep domain adaptation can be mainly grouped into two categories: *1) moment matching*. These methods align representations across domains by minimizing the discrepancy between feature distributions [51, 29, 31, 32, 28]; *2) adversarial training*. These methods adversarially learn transferable feature representations by confusing a domain discriminator in a two-player game [14, 50, 30, 55, 60].

While numerous domain adaptation methods have been proposed, most of them mainly focus on improving the accuracy in the target domain but fail to estimate the predictive uncertainty, falling

---

[*]Corresponding author: Mingsheng Long (mingsheng@tsinghua.edu.cn)

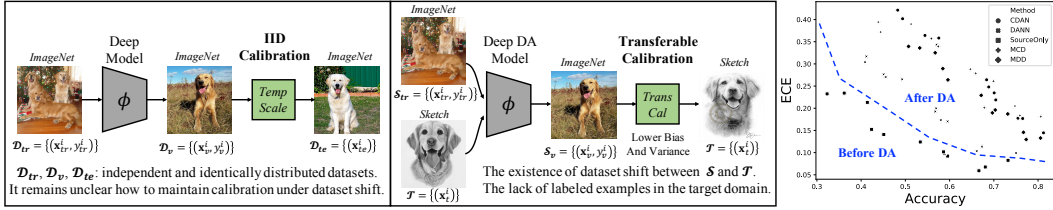

Figure 1: **Left**: A comparison between IID Calibration with TransCal, where $\phi$ denotes the deep model; **Right**: an observation on the accuracy and ECE of various DA methods (12 transfer tasks of Office-Home [52] with ResNet-50 [22]), indicating that DA models learn higher accuracy than the SourceOnly ones *at the expense of* well-calibrated probabilities. See more results in D.1 of *Appendix*.

short of a miscalibration problem [20]. The accuracy of a deep adapted model constitutes only one side of the coin, here we delve into the other side of the coin, i.e. *the calibration of accuracy and confidence*, which requires the model to output a probability that reflects the true frequency of an event. For example, if an automated diagnosis system says 1,000 patients have lung cancer with probability 0.1, approximately 100 of them should indeed have lung cancer. Calibration is fundamental to deep neural models and of great significance for decision-making in safety-critical scenarios. With built-in [12, 25] or post-hoc [42, 20] recalibration methods, the confidence and accuracy of deep models can be well-calibrated in the independent and identically distributed (IID) scenarios. However, it remains unclear how to maintain calibration under dataset shifts, especially when we do not have labels from the target dataset, as in the general setting of Unsupervised Domain Adaptation (UDA). We identify two obstacles in the way of applying calibration to UDA:

- *The lack of labeled examples in the target domain.* We know that the existing successful post-hoc IID recalibration methods mostly rely on ground-truth labels in the validation set to select the optimal temperature [42, 20]. However, since ground-truth labels are not available in the target domain, it is not feasible to directly apply IID calibration methods to UDA.

- *Dataset shift entangled with the miscalibration of DNNs.* Since DNNs are believed to learn more transferable features [35, 57], many domain adaptation methods embed DNNs to implicitly close the domain shift and rely on DNNs to achieve higher classification accuracy. However, DNNs are prone to over-confidence [20], falling short of a miscalibration problem.

To this end, we study the open problem of *Calibration in DA*, which is extremely challenging due to the coexistence of the domain gap and the lack of target labels. To figure out the calibration error on the target domain of DA models, we first delve into the predictions and confidences of the target dataset. By calculating the target accuracy and ECE [20] (a calibration error measure defined in 3.1) with various domain adaptation models before calibration, we found something interesting. As shown in the right panel of Figure 1, the accuracy increases from the weakest SourceOnly [22] model to the latest state-of-the-art MDD [60] model, while the ECE becomes larger as well. That is, after applying domain adaptation methods, miscalibration phenomena become severer compared with SourceOnly model, indicating that the domain adaptation models learn higher classification accuracy *at the expense of* well-calibrated probabilities. This dilemma is unacceptable in safety-critical scenarios, as we need higher accuracy while maintaining calibration. Worse still, the well-performed calibration methods in the IID setting cannot be directly applied to DA due to the domain shift.

To tackle the dilemma between accuracy and calibration, we propose a new Transferable Calibration (TransCal) method in DA, achieving more accurate calibration with lower bias and variance in a unified hyperparameter-free optimization framework, while a comparison with IID calibration is shown in the left panel of Figure 1. Specifically, we first define a new calibration measure, *Importance Weighted Expected Calibration Error* (IWECE) to estimate the calibration error in the target domain in a transferable calibration framework. Next, we propose a *learnable meta parameter* to further reduce the estimation bias from the perspective of theoretical analysis. Meanwhile, we develop a *serial control variate* method to further reduce the variance of the estimated calibration error. As a general post-hoc calibration method, TransCal can be easily applied to recalibrate existing DA methods. This paper has the following contributions:

- We uncover a dilemma in the open problem of Calibration in DA: existing domain adaptation models learn higher classification accuracy *at the expense of* well-calibrated probabilities.

- We propose a Transferable Calibration (TransCal) method, achieving more accurate calibration with lower bias and variance in a unified hyperparameter-free optimization framework.
- We conduct extensive experiments on various DA methods, datasets, and calibration metrics, while the effectiveness of our method has been justified both theoretically and empirically.

## 2 Related Work

### 2.1 Domain Adaptation

There are mainly two types of domain adaptation formulas: *covariate shift* [44, 37, 29, 13] and *label shift* [27, 2, 1], while we focus on the former in this paper since it appears more natural in recognition tasks and attracts more attention in the literature. Existing domain adaptation methods can be mainly grouped into two categories: *moment matching* and *adversarial training*. Moment matching methods align feature distributions across domains by minimizing the distribution discrepancy, in which Maximum Mean Discrepancy [19] is adopted by DAN [29] and DDC [51], and Joint Maximum Mean Discrepancy is utilized by JAN [32]. Motivated by Generative Adversarial Networks (GAN) [17], DANN [14] introduces a domain discriminator to distinguish the source features from the target ones, which are generated by the feature extractor. The domain discriminator and feature extractor are competing in a two-player minimax game. Further, CDAN [30] conditions the adversarial domain adaptation models on discriminative information conveyed in the classifier predictions. MADA [39] uses multiple domain discriminators to capture multimodal structures for fine-grained domain alignment. ADDA [50] adopts asymmetric feature extractors while MCD [47] employs two classifiers consistent across domains. MDD [60] proposes a new domain adaptation margin theory and achieves an impressive performance. TransNorm [54] tackles domain adaptation from a new perspective of designing a transferable normalization layer. Though numerous DA methods have been proposed, most of them focus on improving target accuracy and rare attention has been paid to the predictive uncertainty, causing a miscalibration between accuracy and confidence.

Table 1: Comparisons among calibration methods for unsupervised domain adaptation (UDA).

| Calibration Method | works with domain shift | works without target label | Bias Reduction | Variance Reduction |
|---|---|---|---|---|
| **Temp. Scaling** [20] | ✗ | ✗ | ✗ | ✗ |
| **MC-dropout** [12] | ✓ | ✗ | ✗ | ✗ |
| **CPCS** [38] | ✓ | ✓ | ✗ | ✗ |
| **TransCal** (proposed) | ✓ | ✓ | ✓ | ✓ |

### 2.2 Calibration

Among *binary* calibration methods, Histogram Binning [58] is a simple non-parametric one with either equal-width or equal-frequency bins; Isotonic Regression [59] is a strict generalization of histogram binning by jointly optimizing the bin boundaries and bin predictions; Differently, Platt Scaling [42] is a parametric one that transforms the logits of a classifier to probabilities. When extended to *multiclass*, there are two types of methods. *1) built-in methods*: Monte Carlo dropout (MC-dropout) [12] is popular as it simply uses Dropout [48] during testing phase to estimate predictive uncertainty. Later, [25] finds out that the ensembles of neural networks can work. Further, Stochastic Variational Bayesian Inference (SVI) methods for deep learning [5, 33, 56] are shown effective. However, built-in methods require to modify the classifier learning algorithm or training procedure, which are complex to apply in DA. Thus, we prefer *2) post-hoc approaches*, including various multi-class extensions of Platt scaling [42]: matrix scaling, vector scaling and temperature scaling [20]. Though remarkable advances of IID calibration are witnessed, it remains unclear how to maintain calibration under dataset shifts [24], especially when the target labels are unavailable in UDA case. Recently, [36] finds that traditional post-hoc IID recalibration methods such as temperature scaling fail to maintain calibration under distributional shift. A recent paper (CPCS) [38] considering calibration under dataset shift uses importance weighting to correct for the shift from the source to the target distribution and applies domain adaptation as a base tool for alignment, while we focus on how to maintain calibration in DA. A detailed comparison of typical calibration methods is shown in Table 1.

# 3 Approach

Let $\mathbf{x}$ denote the input of the network, $\mathbf{y}$ be the label and $d$ be the Bernoulli variable indicating to which domain $\mathbf{x}$ belongs. In our terminology, the source domain distribution is $p(\mathbf{x})$ and the target domain distribution is $q(\mathbf{x})$. We are given a labeled source domain $\mathcal{S} = \left\{ \left( \mathbf{x}_s^i, \mathbf{y}_s^i \right) \right\}_{i=1}^{n_s}$ with $n_s$ samples ($d = 1$), and an unlabeled target domain $\mathcal{T} = \left\{ \left( \mathbf{x}_t^i \right) \right\}_{i=1}^{n_t}$ with $n_t$ samples ($d = 0$). Similar to IID calibration, $\mathcal{S}$ is first partitioned into $\mathcal{S}_{tr} = \left\{ \left( \mathbf{x}_{tr}^i, \mathbf{y}_{tr}^i \right) \right\}_{i=1}^{n_{tr}}$ and $\mathcal{S}_v = \left\{ \left( \mathbf{x}_v^i, \mathbf{y}_v^i \right) \right\}_{i=1}^{n_v}$. In this paper, we proposed a new Transferable Calibration (TransCal) method in DA under the well-known covariate shift assumption, *i.e.*, the equation $p(\mathbf{y}|\mathbf{x}) = q(\mathbf{y}|\mathbf{x})$ is held.

## 3.1 IID Calibration

**Calibration Metrics.** Given a deep neural model $\phi$ (parameterized by $\theta$) which transforms the random variable input $X$ into the class prediction $\widehat{Y}$ and its associated confidence $\widehat{P}$, we can define the *perfect calibration* [20] as $\mathbb{P}(\widehat{Y} = Y | \widehat{P} = c) = c, \ \forall \ c \in [0, 1]$ where $Y$ is the ground truth label. There are some typical metrics to measure calibration error: **1)** Negative Log-Likelihood (NLL) [18], also known as the cross-entropy loss in field of deep learning, serves as a proper scoring rule to measure the quality of a probabilistic model [21]. **2)** Brier Score (BS) [6], defined as the squared error between $p(\widehat{\mathbf{y}}|\mathbf{x}, \boldsymbol{\theta})$ and $\mathbf{y}$, is another proper scoring rule for uncertainty measurement. **3)** Expected Calibration Error (ECE) [34, 20] first partitions the interval of probability predictions into $B$ bins where $B_m$ is the indices of samples falling into the $m$-th bin, and then computes the weighted absolute difference between accuracy and confidence across bins:

$$\mathcal{L}_{\text{ECE}} = \sum_{m=1}^{B} \frac{|B_m|}{n} |\mathbb{A}(B_m) - \mathbb{C}(B_m)|, \tag{1}$$

where for each bin $m$, the accuracy is $\mathbb{A}(B_m) = |B_m|^{-1} \sum_{i \in B_m} \mathbf{1}(\widehat{\mathbf{y}}_i = \mathbf{y}_i)$ and its confidence is $\mathbb{C}(B_m) = |B_m|^{-1} \sum_{i \in B_m} \max_k p(\widehat{\mathbf{y}}_i^k | \mathbf{x}_i, \boldsymbol{\theta})$. ECE is easier to interpret and thereby more popular.

**Temperature Scaling Calibration.** Temperature scaling is one of the simplest, fastest, and effective IID Calibration methods [20]. Fixing the neural model trained on the training set $\mathcal{D}_{tr}$, temperate scaling first attains the optimal temperature $T^*$ by minimizing the cross-entropy loss between the logit vectors $\mathbf{z}_v$ scaled by temperatrue $T$ and the ground truth label $\mathbf{y}_v$ on the validation set $\mathcal{D}_v$ as

$$T^* = \arg\min_T \ \mathsf{E}_{(\mathbf{x}_v, \mathbf{y}_v) \in \mathcal{D}_v} \ \mathcal{L}_{\text{NLL}} \left( \sigma(\mathbf{z}_v/T), \mathbf{y}_v \right), \tag{2}$$

where $\sigma(\cdot)$ is the *softmax* function as $\sigma(z_j) = \exp(z_j) / \sum_{k=1}^{K} \exp(z_k)$ for $K$ classes. After that, we transform the logit vectors $\mathbf{z}_{te}$ on the test set $\mathcal{D}_{te}$ into calibrated probabilities by $\widehat{\mathbf{y}}_{te} = \sigma(\mathbf{z}_{te}/T^*)$.

## 3.2 Transferable Calibration Framework

As mentioned above, the main challenge of extending temperature scaling method into domain adaptation (DA) setup is that the target calibration error $\mathbb{E}_q = \mathsf{E}_{\mathbf{x} \sim q} \left[ \mathcal{L}_{(\cdot)}(\phi(\mathbf{x}), \mathbf{y}) \right]$ is defined over the target distribution $q$ where labels are inaccessible. However, if density ratio (a.k.a. importance weight) $w(\mathbf{x}) = q(\mathbf{x})/p(\mathbf{x})$ is known, we can estimate target calibration error by the source distribution $p$:

$$\begin{aligned}
\mathsf{E}_{\mathbf{x} \sim q} \left[ \mathcal{L}_{(\cdot)}(\phi(\mathbf{x}), \mathbf{y}) \right] &= \int_q \mathcal{L}_{(\cdot)}(\phi(\mathbf{x}), \mathbf{y}) q(\mathbf{x}) \mathrm{d}\mathbf{x} \\
&= \int_p \frac{q(\mathbf{x})}{p(\mathbf{x})} \mathcal{L}_{(\cdot)}(\phi(\mathbf{x}), \mathbf{y}) p(\mathbf{x}) \mathrm{d}\mathbf{x} = \mathsf{E}_{\mathbf{x} \sim p} \left[ w(\mathbf{x}) \mathcal{L}_{(\cdot)}(\phi(\mathbf{x}), \mathbf{y}) \right],
\end{aligned} \tag{3}$$

which means $\mathsf{E}_{\mathbf{x} \sim p} \left[ w(\mathbf{x}) \mathcal{L}_{(\cdot)}(\phi(\mathbf{x}), \mathbf{y}) \right]$ is an *unbiased* estimator of the target calibration error $\mathbb{E}_q$. In Eq. (3), it is obvious that there are two buliding blocks: importance weight $w(\mathbf{x})$ and calibration metric $\mathcal{L}_{(\cdot)}$. We first delve into the specific type of calibration metric $\mathcal{L}_{(\cdot)}$. The existing calibration method under covariate shift (CPCS) [38] utilizes the Brier Score $\mathcal{L}_{\text{BS}}$. However, Brier Score conflates accuracy with calibration since it can be decomposed into two components: calibration error and refinement [10], making it insensitive to predicted probabilities associated with infrequent

events [36]. Meanwhile, NLL is minimized if and only if the prediction recovers ground truth $\mathbf{y}$, however, it may over-emphasize tail probabilities [7]. Hence, we adopt ECE, an intuitive and informative calibration metric to directly quantify the goodness of calibration. One may concern that ECE is not a proper scoring rule since the optimum score may not correspond to a perfect prediction, however, as a post-hoc method that softens the overconfident probabilities but *keeps the probability order over classes*, the temperature scaling we utilize will maintain the same accuracy with that before calibration, while achieving a lower ECE as shown in Fig. 2. Since this kind of calibration is trained on the source data but can transfer to the target domain, we call it *transferable calibration*.

Previously, we assume that density ratio is known, however, it is not readily accessible in real-world applications. In this paper, we adopt a mainstream discriminative density ratio estimation method: LogReg [43, 3, 8], which uses Bayesian formula to derive the estimated density ratio from a logistic regression classifier that separates examples from the source and the target domains as

$$\widehat{w}(\mathbf{x}) = \frac{q(\mathbf{x})}{p(\mathbf{x})} = \frac{v(\mathbf{x}|d=0)}{v(\mathbf{x}|d=1)} = \frac{P(d=1)}{P(d=0)} \frac{P(d=0|\mathbf{x})}{P(d=1|\mathbf{x})}, \tag{4}$$

where $v$ is a distribution over $(\mathbf{x}, d) \in \mathcal{X} \times \{0, 1\}$ and $d \sim \mathrm{Bernoulli}(0.5)$ is a Bernoulli variable indicating to which domain $\mathbf{x}$ belongs. With Eq. (4), the estimated density ratio $\widehat{w}(\mathbf{x})$ can be decomposed into two parts, in which the first part $P(d=1)/P(d=0)$ is a constant weight factor that can be estimated with the sample sizes of source and target domains as $n_s/n_t$, and the second part $P(d=0|\mathbf{x})/P(d=1|\mathbf{x})$ is the ratio of target probability to source probability that can be directly estimated with the probabilistic predictions of the logistic regression classifier. For simplicity, we randomly upsample the source or the target dataset to make $n_s = n_t$, *i.e.*, $P(d=1)/P(d=0)$ equals to 1.0. In this way, $\widehat{w}(\mathbf{x})$ is only decided by the second part: $P(d=0|\mathbf{x})/P(d=1|\mathbf{x})$.

### 3.3 Bias Reduction by Learnable Meta Parameter

Through the above analysis, we can reach an *unbiased* estimation of the target calibration error if the estimated importance weights are equal to the true ones. However, the gap between them are non-ignorable, causing a *bias* between the estimated calibration error and the ground-truth calibration error in the target domain. We formalize this bias of calibration as

$$\left| \mathsf{E}_{\mathbf{x} \sim q} \left[ \mathcal{L}_{\mathrm{ECE}}^{\widehat{w}(\mathbf{x})} \right] - \mathsf{E}_{\mathbf{x} \sim q} \left[ \mathcal{L}_{\mathrm{ECE}}^{w(\mathbf{x})} \right] \right| = \left| \mathsf{E}_{\mathbf{x} \sim p} \left[ \widehat{w}(\mathbf{x}) \mathcal{L}_{\mathrm{ECE}}(\phi(\mathbf{x}), \mathbf{y}) \right] - \mathsf{E}_{\mathbf{x} \sim p} \left[ w(\mathbf{x}) \mathcal{L}_{\mathrm{ECE}}(\phi(\mathbf{x}), \mathbf{y}) \right] \right|$$
$$= \left| \mathsf{E}_{\mathbf{x} \sim p} \left[ (w(\mathbf{x}) - \widehat{w}(\mathbf{x})) \mathcal{L}_{\mathrm{ECE}}(\phi(\mathbf{x}), \mathbf{y}) \right] \right|. \tag{5}$$

Note that the bias of estimated calibration error in the target domain is highly related to the estimation error of importance weights. Hence, we focus on the bias of importance weights and show that after applying some basic mathematical inequalities, the estimation bias can be bounded by

$$\left| \mathsf{E}_{\mathbf{x} \sim p} \left[ (w(\mathbf{x}) - \widehat{w}(\mathbf{x})) \mathcal{L}_{\mathrm{ECE}}(\phi(\mathbf{x}), \mathbf{y}) \right] \right|$$
$$\leq \sqrt{\mathsf{E}_{\mathbf{x} \sim p} \left[ (w(\mathbf{x}) - \widehat{w}(\mathbf{x}))^2 \right] \mathsf{E}_{\mathbf{x} \sim p} \left[ (\mathcal{L}_{\mathrm{ECE}}(\phi(\mathbf{x}), \mathbf{y}))^2 \right]} \quad (\mathrm{Cachy - Schwarz\ Ineqaulity})$$
$$\leq \frac{1}{2} \left( \mathsf{E}_{\mathbf{x} \sim p} \left[ (w(\mathbf{x}) - \widehat{w}(\mathbf{x}))^2 \right] + \mathsf{E}_{\mathbf{x} \sim p} \left[ (\mathcal{L}_{\mathrm{ECE}}(\phi(\mathbf{x}), \mathbf{y}))^2 \right] \right) \quad (\mathrm{AM/GM\ Inequality}) \tag{6}$$

where AM/GM denotes the inequality of arithmetic and geometric means. It is noteworthy that the domain adaptation model $\phi$ is fixed since we consider transferable calibration as a post-hoc method. Therefore, we can safely bypass the second term of Eq. (6) and focus our attention on the first term. According to the standard *bounded importance weight assumption* [9], for some bound $M > 0$ we have $w(\mathbf{x}) \leq M$. Then for any $\mathbf{x}$ s.t. $P(d=1|\mathbf{x}) \neq 0$, the following inequality holds:

$$\frac{1}{M+1} \leq P(d=1|\mathbf{x}) \leq 1, \quad \text{since } w(\mathbf{x}) = \frac{P(d=0|\mathbf{x})}{P(d=1|\mathbf{x})} = \frac{1 - P(d=1|\mathbf{x})}{P(d=1|\mathbf{x})} = \frac{1}{P(d=1|\mathbf{x})} - 1. \tag{7}$$

In this way, the first term of the bias in importance weights in Eq. (6) can be further bounded by

$$\mathsf{E}_{\mathbf{x} \sim p} \left[ (w(\mathbf{x}) - \widehat{w}(\mathbf{x}))^2 \right] = \mathsf{E}_{\mathbf{x} \sim p} \left[ \left( \frac{P(d=1|\mathbf{x}) - \widehat{P}(d=1|\mathbf{x})}{P(d=1|\mathbf{x})\widehat{P}(d=1|\mathbf{x})} \right)^2 \right]$$
$$\leq (M+1)^4 \mathsf{E}_{\mathbf{x} \sim p} \left[ \left( P(d=1|\mathbf{x}) - \widehat{P}(d=1|\mathbf{x}) \right)^2 \right]. \tag{8}$$

Plugging Eq. (8) into Eq. (6), we conclude that a smaller $M$ can ensure a lower bias for the estimated weight $\widehat{w}(\mathbf{x})$, leading to a smaller bias of the estimated target calibration error, which is also supported by the generalization bound for importance weighting domain adaptation (Theorem 1, [9]).

To this end, what we should do is to find some techniques to control the upper bound $M$ of importance weights. It seems that we can normalize each weight by the sum of all weights, leading to a smaller $M$. Still, only with self-normalization, a few bad samples with very large weights will dominate the estimation, and drastically explode the estimator. Further, can we clip those samples with very large weights by a given threshold? It seems feasible, but the threshold is task-specific and hard to preset, which is not an elegant solution that we pursue. Based on the above theoretical analysis, we propose to introduce a learnable meta parameter $\lambda$ ($0 \leq \lambda \leq 1$) to adaptively downscale the extremely large weights, which can decrease $M$ and attain a bias-reduced target calibration error. Formally,

$$T^* = \arg\min_{T,\lambda} \mathsf{E}_{\mathbf{x}_v \sim p}\left[\widetilde{w}(\mathbf{x}_v)\mathcal{L}_{\mathrm{ECE}}(\sigma(\phi(\mathbf{x}_v)/T),\ \mathbf{y})\right], \quad \widetilde{w}(\mathbf{x}_v^i) = \left[\widehat{w}(\mathbf{x}_v^i)\right]^\lambda. \qquad (9)$$

By jointly optimizing the calibration objective in Eq. (9), we can attain an optimal temperature $T^*$ for transferable calibration, along with a task-specific optimal $\lambda^*$ for bias reduction. [49] also introduced a control value to importance weighting for model selection, but it was used as a hyperparameter. This work further makes itself learnable in a unified hyperparameter-free optimization framework.

### 3.4 Variance Reduction by Serial Control Variate

Through the above analysis, we enable transferable calibration and further reduce its bias. However, another main drawback of importance weighting is uncontrolled *variance* as the importance weighted estimator can be drastically exploded by a few bad samples with large weights. For simplicity, we denote $w(\mathbf{x})\mathcal{L}_{\mathrm{ECE}}(\phi(\mathbf{x}),\ \mathbf{y})$ as $\mathcal{L}_{\mathrm{ECE}}^w$. Replacing the estimated target error from Lemma 2 of [9] with $\mathcal{L}_{\mathrm{ECE}}^w$, we can conclude that the variance of transferable calibration error can be bounded by Rényi divergence between $p$ and $q$ (A proof is provided in B.1 of *Appendix*):

$$\begin{aligned}
\mathrm{Var}_{\mathbf{x}\sim p}\left[\mathcal{L}_{\mathrm{ECE}}^w\right] &= \mathsf{E}_{\mathbf{x}\sim p}\left[(\mathcal{L}_{\mathrm{ECE}}^w)^2\right] - (\mathsf{E}_{\mathbf{x}\sim p}\left[\mathcal{L}_{\mathrm{ECE}}^w\right])^2 \\
&\leq d_{\alpha+1}(q\|p)(\mathsf{E}_{\mathbf{x}\sim p}\mathcal{L}_{\mathrm{ECE}}^w)^{1-\frac{1}{\alpha}} - (\mathsf{E}_{\mathbf{x}\sim p}\mathcal{L}_{\mathrm{ECE}}^w)^2, \quad \forall \alpha > 0.
\end{aligned} \qquad (10)$$

Apparently, lowering the variance of $\mathcal{L}_{\mathrm{ECE}}^w$ results in more accurate estimation. First, Rényi divergence [45] between $p$ and $q$ can be reduced by deep domain adaptation methods [30, 14, 60]. Second, developing bias reduction term in 3.3 may unexpectedly increase the estimation variance, thus we further reduce the variance via the *parameter-free* control variate method [26]. It introduces a related unbiased estimator $t$ to the estimator $u$ that we concern, achieving a new estimator $u^* = u + \eta(t - \tau)$ while $\mathbb{E}[t] = \tau$. As proved in A.3 of *Appendix*, $\mathrm{Var}[u^*] \leq \mathrm{Var}[u]$ is held and $u^*$ has an optimal solution when $\hat{\eta} = -\mathrm{Cov}(u, t)/\mathrm{Var}[t]$. For brevity, denote $\widetilde{\mathbb{E}}_q(\widehat{\mathbf{y}}, \mathbf{y}) = \mathsf{E}_{\mathbf{x}\sim p}\left[\widetilde{w}(\mathbf{x})\mathcal{L}_{\mathrm{ECE}}(\phi(\mathbf{x}),\ \mathbf{y})\right]$ as $\mathsf{E}_{\mathbf{x}\sim p}\mathcal{L}_{\mathrm{ECE}}^{\widetilde{w}}$ hereafter. To reduce $\mathrm{Var}_{\mathbf{x}\sim p}[\mathcal{L}_{\mathrm{ECE}}^{\widetilde{w}}]$, we first adopt the importance weight $\widetilde{w}(\mathbf{x})$ as the control variate since the expectation of $\widetilde{w}(\mathbf{x})$ is approximately fixed: $\mathsf{E}_{\mathbf{x}\sim p}\left[\widetilde{w}(\mathbf{x})\right] = 1$. Here, regard $\mathsf{E}_{\mathbf{x}\sim p}[\mathcal{L}_{\mathrm{ECE}}^{\widetilde{w}}]$ and $\widetilde{w}(\mathbf{x})$ as $u$ and $t$ respectively, and we can attain a new unbiased estimator. When $\eta$ achieves the optimal solution, the estimation of target calibration error with control variate is

$$\mathbb{E}_q^*(\widehat{\mathbf{y}}, \mathbf{y}) = \widetilde{\mathbb{E}}_q(\widehat{\mathbf{y}}, \mathbf{y}) - \frac{1}{n_s}\frac{\mathrm{Cov}(\mathcal{L}_{\mathrm{ECE}}^{\widetilde{w}}, \widetilde{w}(\mathbf{x}))}{\mathrm{Var}[\widetilde{w}(\mathbf{x})]}\sum_{i=1}^{n_s}\left[\widetilde{w}(\mathbf{x}_s^i) - 1\right]. \qquad (11)$$

Further, we can add the prediction correctness on the source domain $r(\mathbf{x}) = \mathbf{1}(\widehat{\mathbf{y}} = \mathbf{y})$ as another control variate because its expectation is also fixed: $\mathsf{E}_{\mathbf{x}\sim p}\left[r(\mathbf{x})\right] = c$, *i.e.*, the accuracy should be equal to the confidence $c$ on a perfect calibrated source model as defined in Section 3.1. In this way, control variate method can be easily extended into the serial version in which there is a collection of control variables: $t_1, t_2$ whose corresponding expectations are $\tau_1, \tau_2$ respectively. Formally,

$$\begin{aligned}
u^* &= u + \eta_1(t_1 - \tau_1), \\
u^{**} &= u^* + \eta_2(t_2 - \tau_2).
\end{aligned} \qquad (12)$$

Plugging $r(\mathbf{x})$ as the second control variate into the bottom line of Eq. (12), we can further reduce the variance of target calibration error by the serial control variate method as

$$\mathbb{E}_q^{**}(\widehat{\mathbf{y}}, \mathbf{y}) = \mathbb{E}_q^*(\widehat{\mathbf{y}}, \mathbf{y}) - \frac{1}{n_s}\frac{\mathrm{Cov}(\mathcal{L}_{\mathrm{ECE}}^{\widetilde{w}*}, r(\mathbf{x}))}{\mathrm{Var}[r(\mathbf{x})]}\sum_{i=1}^{n_s}\left[r(\mathbf{x}_s^i) - c\right], \qquad (13)$$

where $\mathcal{L}_{\mathrm{ECE}}^{\widetilde{w}*}$ is the estimated target calibration error after applying the control variate to weight $\widetilde{w}(\mathbf{x})$. Similarly, replacing the $\mathsf{E}_{\mathbf{x}\sim p}\mathcal{L}_{\mathrm{ECE}}^{\widetilde{w}}$ defined in Eq. 9 with $\mathbb{E}_q^{**}(\widehat{\mathbf{y}}, \mathbf{y})$ defined in Eq. 13 and then optimizing the new objective, we can attain a more accurate calibration with lower bias and variance.

---

**Algorithm 1** Transferable Calibration in Domain Adaptation

---

1: **Input:** Labeled source dataset $\mathcal{S} = \left\{ \left( \mathbf{x}_s^i, \mathbf{y}_s^i \right) \right\}_{i=1}^{n_s}$ and unlabled target dataset $\mathcal{T} = \left\{ \left( \mathbf{x}_t^i \right) \right\}_{i=1}^{n_t}$

2: **Parameter:** Temperature $T$ and learnable meta parameter $\lambda$

3: Partition $\mathcal{S}$ into $\mathcal{S}_{tr} = \left\{ \left( \mathbf{x}_{tr}^i, \mathbf{y}_{tr}^i \right) \right\}_{i=1}^{n_{tr}}$ and $\mathcal{S}_v = \left\{ \left( \mathbf{x}_v^i, \mathbf{y}_v^i \right) \right\}_{i=1}^{n_v}$

4: Train a DA model $\phi(\mathbf{x}) = G(F(\mathbf{x}))$ on $\mathcal{S}_{tr}$ and $\mathcal{T}$ via any DA method until convergy

5: Randomly upsample the source or the target dataset to make $n_{tr} = n_t$

6: Fix the DA model and compute features $\mathcal{F}_{tr} = \left\{ \boldsymbol{f}_{tr}^i \right\}_{i=1}^{n_{tr}}, \mathcal{F}_v = \left\{ \boldsymbol{f}_v^i \right\}_{i=1}^{n_v}, \mathcal{F}_t = \left\{ \boldsymbol{f}_t^i \right\}_{i=1}^{n_t}$

7: Train a logistic regression model $H$ to discriminate the features $\mathcal{F}_{tr}$ and $\mathcal{F}_t$ until converge

8: Compute $\widehat{w}(\mathbf{x}_v^i) = \left[ 1 - H(\boldsymbol{f}_v^i) \right] / H(\boldsymbol{f}_v^i)$ and $\widetilde{w}(\mathbf{x}_v^i) = \left[ \widehat{w}(\mathbf{x}_v^i) \right]^{\lambda}$

9: Compute $\mathsf{E}_{\mathbf{x} \sim p} \mathcal{L}_{\mathrm{ECE}}^{\widetilde{w}}, \mathbb{E}_q^*(\widehat{\mathbf{y}}, \mathbf{y})$ and $\mathbb{E}_q^{**}(\widehat{\mathbf{y}}, \mathbf{y})$ as in Eq. 9, Eq. 11 and Eq. 13 respectively

10: Jointly optimize the transferable calibration objective as $T^* = \underset{T,\lambda}{\arg\min} \, \mathbb{E}_q^{**} \left( \sigma(\phi(\mathbf{x}_v)/T), \, \mathbf{y}_v \right)$

11: Calibrate the logit vectors on the target domain by $\widehat{\mathbf{y}}_t = \sigma(\phi(\mathbf{x}_t)/T^*)$

---

In summary, the transferable calibration framework (3)–(4) is improved through: 1) *lowering bias* as (9); 2) *lowering variance* by deep adaptation as (10) and by serial control variate as (11) and (13). The overall process of TransCal is summarized in Algorithm 1. Integrating the above explanation, TransCal is designed to achieve more accurate calibration in domain adaptation with lower bias and variance in a unified hyperparameter-free optimization framework.

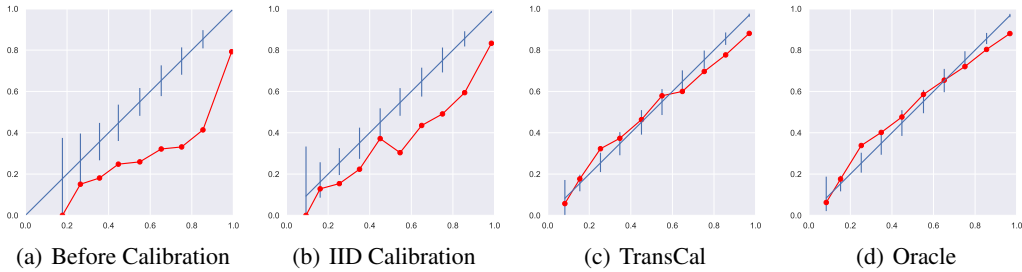

(a) Before Calibration  (b) IID Calibration  (c) TransCal  (d) Oracle

Figure 2: Reliability diagrams from *Clipart* to *Product* with CDAN [30] before and after calibration.

## 4 Experiments

### 4.1 Setup

We fully verify our methods on six DA datasets: (1) *Office-Home* [52]: a dataset with 65 categories, consisting of 4 domains: *Artistic* (**A**), *Clipart* (**C**), *Product* (**P**) and *Real-World* (**R**). (2) *VisDA-2017* [41], a **S**imulation-to-**R**eal dataset with 12 categories. (3) *ImageNet-Sketch* [53], a large-scale dataset transferring from ImageNet (**I**) to Sketch (**S**) with 1000 categories. (4) *Multi-Domain Sentiment* [4], a NLP dataset, comprising of product reviews from *amazon.com* in four product domains: books (**B**), dvds (**D**), electronics (**E**), and kitchen appliances (**K**). (5) *DomainNet* [40]: a dataset with 345 categories, including 6 domains: *Infograph* (**I**), *Quickdraw* (**Q**), *Real* (**R**), *Sketch* (**S**), *Clipart* (**C**) and *Painting* (**P**). (6) *Office-31* [46] contains 31 categories from 3 domains: *Amazon* (**A**), *Webcam* (**W**), *DSLR* (**D**). We run each experiment for 10 times. We denote *Vanilla* as the standard softmax method before calibration, *Oracle* as the temperature scaling method while the target labels are available. Detailed descriptions are included in C.1, C.2 and C.3 of *Appendix*.

### 4.2 Results

**Qualitative Results.** As shown in Figure 2, the blue lines indicate the distributions for *perfectly reliable* forecasts with standard deviation, and the red lines denote the conditional distributions of the observations. Obviously, If the model is perfectly calibrated, these two lines should be matched. We

Table 2: ECE (%) vs. Acc (%) via various calibration methods on *Office-Home* with CDAN

| Metric | Cal. Method | A→C | A→P | A→R | C→A | C→P | C→R | R→A | R→C | R→P | Avg |
|---|---|---|---|---|---|---|---|---|---|---|---|
| Acc | Before Cal. | 49.4 | 68.4 | 75.5 | 57.6 | 70.1 | 70.4 | 68.9 | 54.4 | 81.2 | **68.3** |
|  | MC-dropout [12] | 47.2 | 66.2 | 71.4 | 57.1 | 65.7 | 70.6 | 68.3 | 53.6 | 80.7 | 66.7 |
|  | TransCal (ours) | 49.4 | 68.4 | 75.5 | 57.6 | 70.1 | 70.4 | 68.9 | 54.4 | 81.2 | **68.3** |
| ECE | Before Cal. | 40.2 | 26.4 | 17.8 | 35.8 | 23.5 | 21.9 | 24.8 | 36.4 | 14.5 | 26.8 |
|  | MC-dropout [12] | 33.1 | 21.3 | 15.0 | 24.2 | 20.5 | 13.2 | 25.6 | 14.2 | 22.4 | 19.6 |
|  | Matrix Scaling | 44.7 | 28.8 | 19.7 | 36.1 | 25.4 | 24.1 | 38.1 | 15.7 | 29.5 | 29.1 |
|  | Vector Scaling | 34.7 | 18.0 | 11.3 | 23.4 | 15.4 | 11.5 | 27.3 | 8.5 | 20.0 | 18.9 |
|  | Temp. Scaling | 28.3 | 17.6 | 10.1 | **21.2** | 13.2 | 8.2 | 26.0 | 8.8 | 18.1 | 16.8 |
|  | CPCS [38] | 35.0 | 29.4 | 8.3 | 21.3 | 29.0 | **5.6** | **19.9** | 9.1 | 20.3 | 19.8 |
|  | TransCal (w/o Bias) | **21.7** | 10.8 | 5.8 | 27.6 | **9.2** | 6.0 | 27.4 | 5.2 | 16.9 | 14.5 |
|  | TransCal (w/o Variance) | 31.2 | 16.4 | 6.5 | 31.1 | 14.7 | 16.1 | 27.5 | **4.1** | 20.0 | 18.6 |
|  | TransCal (ours) | 22.9 | **9.3** | **5.1** | 21.7 | 14.0 | 6.4 | 21.6 | 4.5 | **15.6** | **13.5** |
|  | Oracle | 5.8 | 8.1 | 4.8 | 10.0 | 7.7 | 4.2 | 5.5 | 3.9 | 6.2 | 6.2 |

Table 3: ECE (%) before and after various calibration methods on several DA methods and datasets.

| Method | Dataset | **Office-Home** | | | | | | | *Sketch* | *VisDA* |
|---|---|---|---|---|---|---|---|---|---|---|
|  | Transfer Task | A→C | A→P | A→R | C→A | C→P | C→R | Avg | I→S | S→R |
| MDD | Before Cal. (Vanilla) | 33.6 | 18.7 | 13.0 | 28.9 | 22.9 | 19.0 | 22.7 | 19.7 | 30.5 |
|  | IID Cal. (Temp. Scaling) | 28.7 | 16.4 | 9.3 | **21.8** | 16.5 | 12.1 | 17.5 | 14.7 | 29.1 |
|  | CPCS [38] | 29.5 | 17.3 | 9.6 | 22.9 | 16.7 | 11.8 | 18.0 | 14.2 | 30.4 |
|  | TransCal (ours) | **13.5** | **11.4** | **4.8** | **21.8** | **7.0** | **11.1** | **11.6** | **8.1** | **16.1** |
|  | Oracle | 6.8 | 8.5 | 4.7 | 7.0 | 5.8 | 4.0 | 6.1 | 4.7 | 7.4 |
| MCD | Before Cal. (Vanilla) | 39.4 | 28.8 | 20.5 | 33.9 | 27.9 | 20.1 | 28.4 | 18.3 | 25.7 |
|  | IID Cal. (Temp. Scaling) | 21.8 | 22.0 | 15.1 | 22.5 | 20.5 | 9.1 | 18.5 | 13.0 | 23.2 |
|  | CPCS [38] | 23.1 | 22.3 | 15.4 | 20.6 | 20.0 | **9.0** | 18.4 | 12.9 | 22.9 |
|  | TransCal (ours) | **13.1** | **20.2** | **5.1** | **15.5** | **9.3** | 9.1 | **12.0** | **10.2** | **7.8** |
|  | Oracle | 5.6 | 9.4 | 2.3 | 7.1 | 7.4 | 2.5 | 5.7 | 3.6 | 1.8 |

can see that TransCal is much better and approaches the *Oracle* one on the task: Clipart → Product. More reliability diagrams of other tasks to back up this conclusion are shown in D.3 of *Appendix*.

**Quantitative Results.** As reported in Table 2 and Table 3, TransCal achieves much lower ECE than competitors (dereases about 30% or more, e.g. when TransCal is used to calibrate MCD on VisDA, the target ECE is reduced from 22.9 to 7.8) on various datasets and domain adaptation methods. Some results of TransCal are even approaching the Oracle ones. Further, the ablation studies on *TransCal (w/o Bias)* and *TransCal (w/o Variance)* verify that both bias reduction term and variance reduction term are effective. TransCal can be generalized to other tasks of Office-Home (D.2.1), to more DA methods (D.2.2), and to DomainNet and Office-31 (D.2.3), all shown in *Appendix*. Further, the results evaluated by NLL and BS metrics are included in D.2.4 and D.2.5 of *Appendix* respectively. Apart from computer vision datasets, TransCal performs well in 12 transfer tasks of a popular NLP dataset: *Amazon Multi-Domain Sentiment* in Table 4. As shown in Table. 2, it is noteworthy that TransCal

Table 4: ECE (%) via various calibration methods on *Multi-Domain Sentiment*.

| Cal. Method | B→D | B→E | B→K | D→B | D→E | D→K | E→B | E→D | E→K | K→B | K→D | K→E | Avg |
|---|---|---|---|---|---|---|---|---|---|---|---|---|---|
| Before Cal. | 13.7 | 15.2 | 17.5 | 20.4 | 18.6 | 21.4 | 11.3 | 10.3 | 23.0 | 13.1 | 14.5 | 20.9 | 16.7 |
| Temp. Scaling | **5.9** | 8.2 | 5.0 | 2.6 | 5.5 | **4.0** | 17.1 | 17.3 | 6.2 | 16.5 | 14.9 | 6.6 | 9.2 |
| TransCal (ours) | 8.0 | **6.1** | **3.8** | **2.4** | **1.4** | **4.0** | **7.7** | **8.4** | **2.2** | **10.9** | **11.2** | **4.2** | **5.9** |
| Oracle | 2.0 | 3.0 | 3.6 | 1.9 | 1.3 | 2.5 | 2.6 | 1.4 | 1.8 | 2.9 | 2.0 | 1.6 | 2.2 |

maintains the same accuracy with that before calibration while built-in methods (*e.g.* MC-dropout) may *degrade* prediction accuracy, and they have to modify the network architecture (*e.g.* adding dropout layers). We further show that both Vector Scaling and Matrix Scaling underperform TransCal and Temp Scaling. Matrix Scaling works even worse than the Vanilla model due to overfitting, which was also observed in the results of Guo *et al.* [20] reported in Table 2.

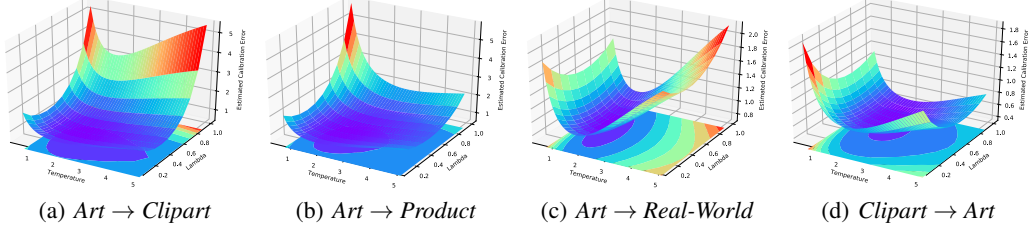

|(a) *Art → Clipart*|(b) *Art → Product*|(c) *Art → Real-World*|(d) *Clipart → Art*|

Figure 3: The estimated calibration error with respect to different values of temperature $T$ and meta parameter $\lambda$ (both are *learnable*), showing that different models achieve optimal values at different $\lambda$.

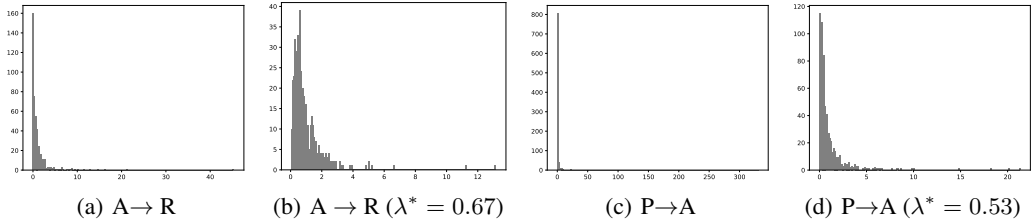

|(a) A→ R|(b) A → R ($\lambda^* = 0.67$)|(c) P→A|(d) P→A ($\lambda^* = 0.53$)|

Figure 4: Importance weight distribution of two DA tasks after transferable calibration with (4(b), 4(d)) and without (4(a), 4(c)) applying the learnable meta parameter, which lowers the value of $M$.

Table 5: ECE (%) of TransCal with different control variate (CV) methods on MDD [60].

| Dataset | *Office-Home* | | | *Sketch* | *VisDA* |
|---|---|---|---|---|---|
| Transfer Task | A→C | A→P | A→R | I→S | S→R |
| TransCal (w/o Control Variate) | 20.9±4.68 | 12.1±2.46 | 6.8±2.22 | 9.7±3.17 | 17.2±5.74 |
| TransCal (CV via only $w(\mathbf{x})$) | 13.9±4.45 | **9.6** ±1.52 | 5.9±1.91 | 9.3±1.68 | 16.4±5.68 |
| TransCal (CV via only $r(\mathbf{x})$) | 13.8±4.32 | 10.2±0.97 | 5.2±1.08 | 8.6±1.37 | 16.3±3.32 |
| TransCal (Parallel Control Variate) | 13.6±4.43 | 10.6±1.46 | 5.2±1.45 | 8.7±1.54 | 16.3±3.45 |
| TransCal (Serial Control Variate) | **13.5±3.51** | 11.4±**0.81** | **4.8±0.76** | **8.1±1.09** | **16.1±1.20** |

### 4.3 Insight Analyses

**Why Bias Reduction Term Works.** From the perspective of optimization, we explore the estimated calibration error with respect to different values of temperature ($T$) and lambda ($\lambda$) in Figure 3, showing that different models achieve optimal values at different $\lambda$. Thus, it is impossible to attain optimal estimated calibration error by presetting a fixed $\lambda$. However, with our unified meta-parameter optimization framework, we can adaptively find an optimal $\lambda$ for each task. From the perspective of importance weight distribution as shown in Figure 4, after applying learnable meta parameter $\lambda$, the highest values ($M$ in Section 3.3) of importance weight decrease, leading to a smaller bias in Eq. (5).

**Why Serial Control Variate Works.** As the theoretical analysis in B.2 of *Appendix* shows, the variance of $\mathbb{E}_q^{**}$ can be further reduced since $\mathrm{Var}[\mathbb{E}_q^{**}] \leq \mathrm{Var}[\mathbb{E}_q^{*}] \leq \mathrm{Var}[\widetilde{\mathbb{E}}_q]$, but other variants of control variate (CV) method such as Parallel CV may not hold this property. Meanwhile, as shown in Table 5, TransCal (Serial CV) not only achieves better calibration performance but also attains lower calibration variance than other variants of control variate methods.

## 5 Conclusion

In this paper, we delve into an open and important problem of *Calibration in DA*. We first reveal that domain adaptation models learn higher accuracy *at the expense of* well-calibrated probabilities. Further, we propose a novel transferable calibration (TransCal) approach, achieving more accurate calibration with lower bias and variance in a unified hyperparameter-free optimization framework.

## Broader Impact

The open problem of *Calibration in DA* that we delve into is a very promising research direction and important for decision making in safety-critical applications, such as automated diagnosis system for lung cancer. Since our method can be easily applied to recalibrate the existing DA methods and generate more reliable predictions, it will benefit the transfer learning community. If the method fails in some extreme circumstances, it will confuse researchers or engineers who apply our method but it will not bring about any negative ethical or societal consequences. Meanwhile, our method did not leverage biases in the data such as racial discrimination and gender discrimination since we conduct experiments on standard domain adaptation datasets that are more about animals or pieces of equipment in the office. In summary, we hold a positive view of the broader impact on this paper.

## Acknowledgements

This work was supported by the National Natural Science Foundation of China (61772299, 71690231), Beijing Nova Program (Z201100006820041), University S&T Innovation Plan by the Ministry of Education of China.

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
