[Supplementary Material]

# Supplementary Material For:
# Transferable Calibration with Lower Bias and Variance in Domain Adaptation

**Ximei Wang, Mingsheng Long,**[*] **Jianmin Wang, and Michael I. Jordan**[♯]
School of Software, KLiss, BNRist, Tsinghua University ♯University of California, Berkeley
wxm17@mails.tsinghua.edu.cn {mingsheng,jimwang}@tsinghua.edu.cn
jordan@cs.berkeley.edu

In this appendix, we will show more explanations, details, and results that are not included in the main paper. In Preliminaries A, we especially add more detailed formal denifitions of calibration metrics, Rényi Divergence, and control variate method. In Proof B, both the bound for the variance of estimated calibration error and the variance reduction analysis for variants of control variate methods are provided. In Seup C, we provide a detailed description of the experiment setting. In the last section D, more qualitative and quantitative results of TransCal are shown when it is evaluated on other domain adaptation tasks of Office-Home, on more domain adaptation methods, on more domain adaptation datasets (*Office-31* and *DomainNet*), and on NLL and BS.

## A  Preliminaries

### A.1  Calibration Metrics.

Given a deep neural model $\phi$ (parameterized by $\theta$) which transforms the random variable input $X$ into the class prediction $\widehat{Y}$ and its associated confidence $\widehat{P}$, we can define the *perfect calibration* [9] as $\mathbb{P}(\widehat{Y} = Y | \widehat{P} = c) = c, \ \forall \ c \in [0, 1]$ where $Y$ is the ground truth label. As it is impossible to achieve perfect calibration in practical, there are some typical metrics to measure calibration error: Negative Log-Likelihood (NLL), Brier Score (BS), and Expected Calibration Error (ECE).

**Negative Log-Likelihood (NLL)** [8], also known as the cross-entropy loss in field of deep learning, serves as a proper scoring rule to measure the quality of a probabilistic model [10]. Denote $p(\widehat{\mathbf{y}}_i | \mathbf{x}_i, \boldsymbol{\theta})$ a predicted probability vector associated with the one-hot encoded ground-truth label $\mathbf{y}_i$ for example $\mathbf{x}_i$ in the dataset, NLL can be defined as

$$\mathcal{L}_{\text{NLL}} = -\sum_{i=1}^{n} \sum_{k=1}^{K} \mathbf{y}_i^k \log p(\widehat{\mathbf{y}}_i^k | \mathbf{x}_i, \boldsymbol{\theta}), \tag{1}$$

where $n$ is the number of samples and $K$ is the number of classes. NLL achieves minimal if and only if the prediction probability $p(\mathbf{y}|\mathbf{x}, \boldsymbol{\theta})$ recovers the ground-truth label $\mathbf{y}$, however, it may over-emphasize tail probabilities [3].

**Brier Score (BS)** [2], defined as the squared error between the prediction probability $p(\mathbf{y}|\mathbf{x}, \boldsymbol{\theta})$ and the ground-truth label $\mathbf{y}$, is another proper scoring rule for uncertainty measurement. Using the same notation of NLL, Brier Score is formally defined as

$$\mathcal{L}_{\text{BS}} = -\frac{1}{K} \sum_{i=1}^{n} \sum_{k=1}^{K} (p(\widehat{\mathbf{y}}_i^k | \mathbf{x}_i, \boldsymbol{\theta}) - \mathbf{y}_i^k)^2, \tag{2}$$

For classification, BS can be decomposed into calibration and refinement [6], therefore, it conflates accuracy with calibration, causing it not a optimal metric for calibration in DA.

---

[*]Corresponding author: Mingsheng Long (mingsheng@tsinghua.edu.cn)

**Expected Calibration Error (ECE)** [16, 9] first partitions the interval of probability predictions into $B$ bins where $B_m$ is the indices of samples falling into the $m$-th bin, and then computes the weighted absolute difference between accuracy and confidence across bins:

$$\mathcal{L}_{\text{ECE}} = \sum_{m=1}^{B} \frac{|B_m|}{n} |\mathbb{A}(B_m) - \mathbb{C}(B_m)|, \tag{3}$$

where for each bin $m$, the accuracy is $\mathbb{A}(B_m) = |B_m|^{-1} \sum_{i \in B_m} \mathbf{1}(\widehat{\mathbf{y}}_i = \mathbf{y}_i)$ and its confidence is $\mathbb{C}(B_m) = |B_m|^{-1} \sum_{i \in B_m} \max_k p(\widehat{\mathbf{y}}_i^k | \mathbf{x}_i, \boldsymbol{\theta})$. ECE is easier to interpret and thereby more popular.

## A.2 Rényi Divergence [22]

Akin to [4, 28], our analysis also based on the widely-used notation of Rényi divergence [22], which is an information-theoretical measure directly relevant to the study of importance weighting. Given a hyper-parameter $\alpha \geq 0$ and $\alpha \neq 1$, Rényi divergence between distribution $p$ and $q$ is defined as $D_\alpha(p \| q) = \frac{1}{\alpha - 1} \log_2 \sum_x p(x) \left( \frac{p(x)}{q(x)} \right)^{\alpha - 1}$. Rényi divergence is well-defined: it is non-negative and $D_\alpha(p \| q) = 0$ if and only if $p = q$. Particularly, when $\alpha = 1$, it coincides with Kullback–Leibler divergence, i.e., $\lim_{\alpha \to 1} D_\alpha(p \| q) = KL(p \| q)$. Here, another notation of Rényi divergence is adopted:

$$d_\alpha(p \| q) = 2^{D_\alpha(p \| q)} = \left[ \sum_x \frac{p^\alpha(x)}{q^{\alpha - 1}(x)} \right]^{\frac{1}{\alpha - 1}}. \tag{4}$$

## A.3 Control Variate [12]

To reduce variance, an effective technique typically employed in Monte Carlo methods named as *Control Variate* [12] is introduced here. Denote the statistic $u$ an unbiased estimator of an unknown parameter $\mu$, i.e. $\mathbb{E}[u] = \mu$. To reduce its variance, we introduce a related unbiased estimator $t$ such that $\mathbb{E}[t] = \tau$, in which $\tau$ is the parameter that $t$ tries to estimate. Then, a new estimator $u^*$ with a constant $\eta$ can be constructed as

$$u^* = u + \eta(t - \tau). \tag{5}$$

$u^*$ has two important properties: 1) $u^*$ is still an *unbiased* unbiased estimator of $\mu$ since $\mathsf{E}[u^*] = \mathsf{E}[u] + \eta \mathsf{E}[t - \tau] = \mu + \eta * (\mathsf{E}[t] - \mathsf{E}[\tau]) = \mu$; 2) The variance $\text{Var}[u^*]$ of $u^*$ is *reduced*, i.e., $\text{Var}[u^*] \leq \text{Var}[u]$. That is because the variance of $u^*$ can be decomposed into

$$\text{Var}[u^*] = \text{Var}[u + \eta(t - \tau)] = \text{Var}[t]\eta^2 + 2\text{Cov}(u, t)\eta + \text{Var}[u], \tag{6}$$

which is a quadratic form of $\eta$ and has a optimal solution when $\widehat{\eta} = -\text{Cov}(u, t)/\text{Var}[t]$, resulting in a optimal value $(1 - \rho(u, t)^2)\text{Var}[u]$ where $\rho$ is the correlation coefficient. Obviously, $\rho$ satisfies $0 \leq |\rho| \leq 1$, leading to a lower variance: $\text{Var}[u^*] \leq \text{Var}[u]$.

# B Proof

## B.1 The Bound for the Variance of Estimated Calibration Error

In the main paper, we have mentioned that the main drawback of importance weighting is uncontrolled *variance* as the importance weighted estimator can be drastically exploded by a few bad samples with large weights. For simplicity, we denote $w(\mathbf{x})\mathcal{L}_{\text{ECE}}(\phi(\mathbf{x}), \mathbf{y})$ as $\mathcal{L}_{\text{ECE}}^w$ as in the main paper. Motivated by the Lemma 2 of the learning bounds for importance weighting [4], we show that the variance of transferable calibration error can be bounded by Rényi divergence between $p$ and $q$. By

using Hölder's Inequality, we provide detailed proof here.

$$
\begin{aligned}
\mathrm{Var}_{\mathbf{x}\sim p}[\mathcal{L}_{\mathrm{ECE}}^{w}] &= \mathsf{E}_{\mathbf{x}\sim p}[(\mathcal{L}_{\mathrm{ECE}}^{w})^2] - (\mathsf{E}_{\mathbf{x}\sim p}[\mathcal{L}_{\mathrm{ECE}}^{w}])^2 \\
&= \mathsf{E}_{\mathbf{x}\sim p}\left[(w(\mathbf{x}))^2\,(\mathcal{L}_{\mathrm{ECE}}(\phi(\mathbf{x}),\,\mathbf{y}))^2\right] - (\mathsf{E}_{\mathbf{x}\sim p}[\mathcal{L}_{\mathrm{ECE}}^{w}])^2 \\
&= \sum_{\mathbf{x}} p(\mathbf{x})\left[\frac{q(\mathbf{x})}{p(\mathbf{x})}\right]^2 (\mathcal{L}_{\mathrm{ECE}}(\phi(\mathbf{x}),\,\mathbf{y}))^2 - (\mathsf{E}_{\mathbf{x}\sim p}[\mathcal{L}_{\mathrm{ECE}}^{w}])^2 \\
&= \sum_{\mathbf{x}} (q(\mathbf{x}))^{\frac{1}{\alpha}}\left[\frac{q(\mathbf{x})}{p(\mathbf{x})}\right] (q(\mathbf{x}))^{\frac{\alpha-1}{\alpha}} (\mathcal{L}_{\mathrm{ECE}}(\phi(\mathbf{x}),\,\mathbf{y}))^2 - (\mathsf{E}_{\mathbf{x}\sim p}[\mathcal{L}_{\mathrm{ECE}}^{w}])^2 \\
&\le \left[\sum_{\mathbf{x}} q(\mathbf{x})\left[\frac{q(\mathbf{x})}{p(\mathbf{x})}\right]^{\alpha}\right]^{\frac{1}{\alpha}}\left[\sum_{\mathbf{x}} q(\mathbf{x})(\mathcal{L}_{\mathrm{ECE}}(\phi(\mathbf{x}),\,\mathbf{y}))^{\frac{2\alpha}{\alpha-1}}\right]^{\frac{\alpha-1}{\alpha}} - (\mathsf{E}_{\mathbf{x}\sim p}[\mathcal{L}_{\mathrm{ECE}}^{w}])^2 \\
&= d_{\alpha+1}(q\|p)\left[\sum_{\mathbf{x}} q(\mathbf{x})\mathcal{L}_{\mathrm{ECE}}(\phi(\mathbf{x}),\,\mathbf{y})(\mathcal{L}_{\mathrm{ECE}}(\phi(\mathbf{x}),\,\mathbf{y}))^{\frac{\alpha+1}{\alpha-1}}\right]^{\frac{\alpha-1}{\alpha}} - (\mathsf{E}_{\mathbf{x}\sim p}[\mathcal{L}_{\mathrm{ECE}}^{w}])^2 \\
&\le d_{\alpha+1}(q\|p)(\mathsf{E}_{\mathbf{x}\sim p}\mathcal{L}_{\mathrm{ECE}}^{w})^{1-\frac{1}{\alpha}}\left[\sum_{\mathbf{x}}\mathcal{L}_{\mathrm{ECE}}(\phi(\mathbf{x}),\,\mathbf{y})\right]^{1+\frac{1}{\alpha}} - (\mathsf{E}_{\mathbf{x}\sim p}[\mathcal{L}_{\mathrm{ECE}}^{w}])^2 \\
&\le d_{\alpha+1}(q\|p)(\mathsf{E}_{\mathbf{x}\sim p}\mathcal{L}_{\mathrm{ECE}}^{w})^{1-\frac{1}{\alpha}} - (\mathsf{E}_{\mathbf{x}\sim p}\mathcal{L}_{\mathrm{ECE}}^{w})^2, \quad \forall\alpha > 0.
\end{aligned}
\tag{7}
$$

Apparently, lowering the variance of $\mathcal{L}_{\mathrm{ECE}}^{w}$ results in a more accurate estimation. First, Rényi divergence [22] between $p$ and $q$ can be reduced by deep domain adaptation methods [14, 7, 29]. Second, we further reduce the variance by the control variate method [12]. As analyzed in the main paper, these two techniques can be utilized to reduce the variance of the transferable calibration error, and the former one has been verified by the previous works. For a fair comparison, we use deep adapted features in all baselines, including the IID Calibration (Temp. Scaling), IID Calibration (Vector Scaling), IID Calibration (Matrix Scaling) and CPCS [17].

## B.2 Variance Reduction Analysis for Variants of Control Variate Methods

### B.2.1 Single Control Variate

As analyzed in Section A.3, control variate is an effective and mainstream technique to reduce variance. By introducing a related unbiased estimator $t$ to the estimator $u$ that we concern, we can attain a new estimator $u^* = u + \eta(t - \tau)$. Obviously, the variance of $u^*$ is

$$
\mathrm{Var}[u^*] = \mathrm{Var}[u + \eta(t - \tau)] = \mathrm{Var}[t]\eta^2 + 2\mathrm{Cov}(u, t)\eta + \mathrm{Var}[u],
\tag{8}
$$

which is a quadratic form of $\eta$ and has a optimal solution when $\widehat{\eta} = -\mathrm{Cov}(u, t)/\mathrm{Var}[t]$, resulting in a optimal value $(1 - \rho(u, t)^2)\mathrm{Var}[u]$ where $\rho$ is the correlation coefficient. Obviously, $\rho$ satisfies $0 \le |\rho| \le 1$, leading to a lower variance: $\mathrm{Var}[u^*] \le \mathrm{Var}[u]$. For a single control variate method, both *Control Variate via only* $w(\mathbf{x})$ as shown in Eq. (9) and *Control Variate via only* $r(\mathbf{x})$ as shown in Eq. (10) can reduce the variance of the target estimated calibration error $\mathrm{Var}[u^*] \le \mathrm{Var}[u]$.

$$
\mathbb{E}_{q}^{(1)}(\widehat{\mathbf{y}}, \mathbf{y}) = \widetilde{\mathbb{E}}_{q}(\widehat{\mathbf{y}}, \mathbf{y}) - \frac{1}{n_s}\frac{\mathrm{Cov}(\mathcal{L}_{\mathrm{ECE}}^{\widetilde{w}}, \widetilde{w}(\mathbf{x}))}{\mathrm{Var}[\widetilde{w}(\mathbf{x})]}\sum_{i=1}^{n_s}[\widetilde{w}(\mathbf{x}_s^i) - 1].
\tag{9}
$$

$$
\mathbb{E}_{q}^{(2)}(\widehat{\mathbf{y}}, \mathbf{y}) = \widetilde{\mathbb{E}}_{q}(\widehat{\mathbf{y}}, \mathbf{y}) - \frac{1}{n_s}\frac{\mathrm{Cov}(\mathcal{L}_{\mathrm{ECE}}^{\widetilde{w}}, r(\mathbf{x}))}{\mathrm{Var}[r(\mathbf{x})]}\sum_{i=1}^{n_s}[r(\mathbf{x}_s^i) - c],
\tag{10}
$$

### B.2.2 Parallel Control Variate

For a parallel control variate method, we extend the control variate method into a parallel version in which there is a collection of control variables: $t_1, t_2$ whose corresponding expectations are $\tau_1, \tau_2$ respectively. By introducing these two related estimators into $u$, a new estimator is attained:

$$
u^* = u + \eta_1(t_1 - \tau_1) + \eta_2(t_2 - \tau_2).
\tag{11}
$$

Similarly, the variance of $u^*$ in the parallel control variate can be decomposed into:

$$
\begin{aligned}
\mathrm{Var}[u^*] &= \mathrm{Var}[u + \eta_1(t_1 - \tau_1) + \eta_2(t_2 - \tau_2)] \\
&= \mathrm{Var}[u] + \mathrm{Var}[t_1]\eta_1^2 + 2\mathrm{Cov}(u, t_1)\eta_1 \\
&\quad + 2\mathrm{Cov}(t_1, t_2)\eta_1\eta_2 + \mathrm{Var}[t_2]\eta_2^2 + 2\mathrm{Cov}(u, t_2)\eta_2,
\end{aligned}
\tag{12}
$$

which is much more complex than that of the single control variate whose variance is a quadratic form and has an optimal solution. Set the derivative of $\mathrm{Var}[u^*]$ with respect to $\eta_1$ and $\eta_2$ to zero:

$$
\begin{aligned}
\mathrm{Var}[t_1]\eta_1 + \mathrm{Cov}(u, t_1) + \mathrm{Cov}(t_1, t_2)\eta_2 = 0 \\
\mathrm{Var}[t_2]\eta_2 + \mathrm{Cov}(u, t_2) + \mathrm{Cov}(t_1, t_2)\eta_1 = 0
\end{aligned}
\tag{13}
$$

we can attain the optimal solutions of $\eta_1$ and $\eta_2$ corresponding to the optimal value of $\mathrm{Var}[u^*]$:

$$
\begin{aligned}
\widehat{\eta}_1 &= \frac{\mathrm{Cov}(u, t_1)\mathrm{Var}[t_2] - \mathrm{Cov}(u, t_2)\mathrm{Cov}(t_1, t_2)}{\mathrm{Cov}(t_1, t_2)\mathrm{Cov}(t_1, t_2) - \mathrm{Var}[t_1]\mathrm{Var}[t_2]} \\
\widehat{\eta}_2 &= \left[\frac{\mathrm{Cov}(u, t_2)\mathrm{Cov}(t_1, t_2) - \mathrm{Cov}(u, t_1)\mathrm{Var}[t_2]}{\mathrm{Cov}(t_1, t_2)\mathrm{Cov}(t_1, t_2) - \mathrm{Var}[t_1]\mathrm{Var}[t_2]}\right] \frac{\mathrm{Var}[t_1]}{\mathrm{Cov}(t_1, t_2)} - \frac{\mathrm{Cov}(u, t_1)}{\mathrm{Cov}(t_1, t_2)}
\end{aligned}
\tag{14}
$$

By pulgging $\widehat{\eta}_1$ and $\widehat{\eta}_2$ into Eq. (12), we can attain the optimal value of $\mathrm{Var}[u^*]$ as $\mathrm{Var}[u]$ + $\mathrm{Res}[t_1, t_2, u]$. However, the property $\mathrm{Var}[u^*] \leq \mathrm{Var}[u]$ is not always true unless we can guarantee that $\mathrm{Res}[t_1, t_2, u] \leq 0$. In this way, the variance of the target estimated calibration error by the parallel control variate method may not be reduced.

### B.2.3 Serial Control Variate

As mentioned in the main paper, the control variate method can be easily extended into the serial version in which there is a collection of control variables: $t_1, t_2$ whose corresponding expectations are $\tau_1, \tau_2$ respectively. That is formally defined as

$$
\begin{aligned}
u^* &= u + \eta_1(t_1 - \tau_1), \\
u^{**} &= u^* + \eta_2(t_2 - \tau_2).
\end{aligned}
\tag{15}
$$

By using the $w(\mathbf{x})$ and $r(\mathbf{x})$ as the first and the second control variate in Eq. (15), we can further reduce the variance of target calibration error by the serial control variate method as

$$
\begin{aligned}
\mathbb{E}_q^*(\widehat{\mathbf{y}}, \mathbf{y}) &= \widetilde{\mathbb{E}}_q(\widehat{\mathbf{y}}, \mathbf{y}) - \frac{1}{n_s} \frac{\mathrm{Cov}(\mathcal{L}_{\mathrm{ECE}}^{\widetilde{w}}, \widetilde{w}(\mathbf{x}))}{\mathrm{Var}[\widetilde{w}(\mathbf{x})]} \sum_{i=1}^{n_s} [\widetilde{w}(\mathbf{x}_s^i) - 1] \\
\mathbb{E}_q^{**}(\widehat{\mathbf{y}}, \mathbf{y}) &= \mathbb{E}_q^*(\widehat{\mathbf{y}}, \mathbf{y}) - \frac{1}{n_s} \frac{\mathrm{Cov}(\mathcal{L}_{\mathrm{ECE}}^{\widetilde{w}*}, r(\mathbf{x}))}{\mathrm{Var}[r(\mathbf{x})]} \sum_{i=1}^{n_s} [r(\mathbf{x}_s^i) - c].
\end{aligned}
\tag{16}
$$

In the serial control variate method, the variance $\mathrm{Var}[u^*]$ and $\mathrm{Var}[u^{**}]$ of $u^*$ and $u^{**}$ are

$$
\begin{aligned}
\mathrm{Var}[u^*] &= \mathrm{Var}[u + \eta_1(t_1 - \tau_1)] = \mathrm{Var}[t_1]\eta_1^2 + 2\mathrm{Cov}(u, t_1)\eta_1 + \mathrm{Var}[u] \\
\mathrm{Var}[u^{**}] &= \mathrm{Var}[u^* + \eta_2(t_2 - \tau_2)] = \mathrm{Var}[t_2]\eta_2^2 + 2\mathrm{Cov}(u^*, t_2)\eta_2 + \mathrm{Var}[u^*].
\end{aligned}
\tag{17}
$$

Apparently, the property $\mathrm{Var}[\mathbb{E}_q^{**}] \leq \mathrm{Var}[\mathbb{E}_q^*] \leq \mathrm{Var}[\widetilde{\mathbb{E}}_q]$ is held since the above two equations in Eq. (17) have optimal solutions when $\widehat{\eta}_1 = -\mathrm{Cov}(u, t_1)/\mathrm{Var}[t_1]$ and $\widehat{\eta}_2 = -\mathrm{Cov}(u^*, t_2)/\mathrm{Var}[t_2]$, resulting in a lower and lower variance of the target estimated calibration error.

## C Setup

### C.1 Datasets

We fully verify our methods on six DA datasets: (1) *Office-Home* [25]: a dataset with 65 categories, consisting of 4 domains: *Artistic* (**A**), *Clipart* (**C**), *Product* (**P**) and *Real-World* (**R**). (2) *VisDA-2017* [19], a **S**imulation-to-**R**eal dataset with 12 categories. (3) *ImageNet-Sketch* [26], a large-scale dataset transferring from ImageNet (**I**) to Sketch (**S**) with 1000 categories. (4) *Multi-Domain Sentiment* [1], a NLP dataset, comprising of product reviews from *amazon.com* in four product domains: books

(**B**), dvds (**D**), electronics (**E**), and kitchen appliances (**K**). (5) *DomainNet* [18]: a dataset with 345 categories, including 6 domains: *Infograph* (**I**), *Quickdraw* (**Q**), *Real* (**R**), *Sketch* (**S**), *Clipart* (**C**) and *Painting* (**P**). (6) *Office-31* [23] contains 31 categories from 3 domains: *Amazon* (**A**), *Webcam* (**W**), *DSLR* (**D**). For each dataset, we randomly split it and use the *first 80 percent* for training and the *remaining 20 percent* data for validation. We run each experiment for 10 times. We denote *Vanilla* as the standard softmax method before calibration, *Oracle* as the temperature scaling method while the target labels are available. Detailed descriptions are included in C.1, C.2 and C.3 of *Appendix*.

## C.2 Implementation Details

Our methods were implemented based on *PyTorch*. The implementation of our paper consists of two main steps: *Generating Features* and *Transferable Calibration*. When generating features, we use ResNet-50 [11] models pre-trained on the ImageNet dataset [21] as the backbone. As a post-hoc calibration method, we fixed the adapted model when recalibrating the accuracy and confidence. As for the Transferable Calibration step, the *scipy.optimize* package was used to solve the constrained optimization problem. Since no hyperparameter was introduced into the method, we can directly attain the results in all experiments. To objectively verify our method, we use three calibration metrics: Negative Log-Likelihood (NLL), Brier Score (BS), and Expected Calibration Error (ECE). Follow the protocol in [9], we set the number of bins $M = 15$ of ECE to measure calibration error. We run each experiment for 10 times for each task.

## C.3 Calibration Methods

We denote *Vanilla* as the standard softmax method before calibration, and *Oracle* as the temperature scaling method while the target labels are available. Meanwhile, *IID Cal. (Temp. Scaling)* is the IID calibration via temperature scaling recalibration method applied on the source domain as adopted in [9], *IID Cal (Platt Scaling)* as the IID calibration via Platt scaling recalibration method adopted in [20]. Further, we report the results of the transferable calibration method *TransCal* that we proposed, and TransCal without bias reduction term: *TransCal (w/o Bias)*, as well as TransCal without variance reduction term: *TransCal (w/o Variance)*. For a fair comparison, we use deep adapted features in all baselines, including the IID Calibration (Temp. Scaling) and CPCS [17]. We select three mainstream domain adaptation methods: MCD [24], CDAN [14] and MDD [29] in the main paper. To verify that TransCal can be generalized to recalibrate domain adaptation models, we further conduct the experiments with the other two mainstream classical domain adaptation methods: DAN [13], JAN [15], and another two latest domain adaptation methods: AFN [27] and BNM [5].

# D Results and Analysis

## D.1 More Results to Demonstrate the Dilemma Between Accuracy and Calibration

In Section 1 of the main paper, we uncover a dilemma in the open problem of Calibration in DA: existing domain adaptation models learn higher classification accuracy *at the expense of* well-calibrated probabilities by 12 transfer tasks of *Office-Home*. To verify these phenomena in other datasets and tasks, we further include the results of accuracy and calibration on *Office-31* with 4 tasks: *Amazon → Webcam*, *Amazon → DSLR*, *DSLR → Amazon*, *Webcam → Amazon* since the other two tasks are too simple, and on *ImageNet-Sketch*, a large-scale dataset transferring from *ImageNet* to *Sketch* consisting of 1000 categories. Note that, besides the five mainstream domain adaptation methods that we reported in the main paper, we further conduct the experiments on other two main stream DA methods: DAN [13], JAN [15], and another latest DA methods: AFN [27] and BNM [5]. As shown in Figure 1, the same conclusion about the dilemma between accuracy and calibration can be drawn on other DA datasets and tasks. Meanwhile, we show the detailed results of accuracy and ECE of 12 transfer tasks on *Office-Home* in Figure 2 to precisely back up our observation of the miscalibration between accuracy and confidence after applying domain adaptation methods.

Figure 1: The dilemma between Accuracy and ECE before calibration on more DA methods and datasets (*Office-Home*, *Office-31*, *Sketch*). After applying domain adaptation methods, miscalibration phenomena become severer compared with SourceOnly model, indicating that DA models learn higher accuracy than the SourceOnly ones *at the expense of* well-calibrated probabilities.

(a) *Art → Clipart*  (b) *Art → Product*  (c) *Art → Real-World*  (d) *Clipart → Art*

(e) *Clipart → Product*  (f) *Clipart → Real-World*  (g) *Product → Art*  (h) *Product → Clipart*

(i) *Product → Real-World*  (j) *Real-World → Art*  (k) *Real-World → Clipart*  (l) *Real-World → Product*

Figure 2: The dilemma between accuracy and ECE for different transfer tasks on *Office-Home*.

## D.2 More Quantitative Results

### D.2.1 Generlized to Other Tasks of *Office-Home*

Due to the space limit, we only report the first six transfer tasks on *Office-Home* in the main paper, thus we show the calibration results of the other tasks in Table 1. As reported, TransCal also achieves much lower ECE than competitors on other tasks on *Office-Home* while recalibrating various domain adaptation methods. Further, the ablation studies on *TransCal (w/o Bias)* and *TransCal (w/o Variance)* also verify that both bias reduction term and variance reduction term are effective.

Table 1: ECE(%) before and after various calibration methods for other 6 tasks on *Office-Home*.

| Method | Transfer Task | P→A | P→C | P→R | R→A | R→C | R→P | Avg |
|---|---|---|---|---|---|---|---|---|
| MDD | Before Cal. (Vanilla) | 26.4 | 33.9 | 14.6 | 19.6 | 32.5 | 13.3 | 23.4 |
| | IID Cal. (Temp. Scaling) | 22.5 | 30.6 | 12.1 | 13.3 | 26.3 | 9.8 | 19.1 |
| | CPCS [17] | 24.6 | 31.6 | 14.1 | 13.3 | 27.0 | 9.9 | 20.1 |
| | TransCal (w/o Bias) | 25.0 | 31.8 | 13.4 | 10.6 | 23.2 | 10.2 | 19.1 |
| | TransCal (w/o Variance) | **21.1** | **29.5** | 12.1 | 12.9 | 24.0 | 9.3 | 18.2 |
| | TransCal (ours) | 21.7 | 30.6 | **6.5** | **7.5** | **23.0** | **5.6** | **15.8** |
| | Oracle | 6.6 | 6.0 | 4.7 | 6.2 | 6.7 | 5.2 | 5.9 |
| MCD | Before Cal. (Vanilla) | 35.7 | 37.2 | 18.4 | 26.1 | 39 | 18.1 | 29.1 |
| | IID Cal. (Temp. Scaling) | 29.1 | 28.1 | 15.9 | 22.6 | 31.1 | 16.3 | 23.9 |
| | CPCS [17] | 30.1 | 30.4 | 15.2 | 21.9 | 32.8 | 17.1 | 24.6 |
| | TransCal (w/o Bias) | 19.1 | **13.7** | 5.9 | 19.3 | 30.7 | 12.4 | 16.8 |
| | TransCal (w/o Variance) | 20.7 | 25.5 | **4.9** | **7.2** | 27.9 | **6.1** | 15.4 |
| | TransCal (ours) | **16.4** | 27.7 | 5.5 | **7.2** | **23.2** | **6.1** | **14.3** |
| | Oracle | 6.2 | 4.7 | 2.6 | 6.9 | 8.1 | 5.3 | 5.6 |
| CDAN | Before Cal. (Vanilla) | 34.2 | 42.1 | 17.7 | 24.8 | 36.4 | 14.5 | 28.3 |
| | IID Cal. (Temp. Scaling) | 25.5 | **32.9** | **11.5** | 14.0 | 26.0 | 8.8 | 19.8 |
| | CPCS [17] | 27.7 | 39.2 | 15.6 | 13.6 | **19.9** | 9.1 | 20.9 |
| | TransCal (w/o Bias) | 26.7 | 38.8 | 13.6 | 10.2 | 27.4 | 5.2 | 20.3 |
| | TransCal (w/o Variance) | 22.1 | 41.7 | 15.7 | 13.0 | 27.5 | **4.1** | 20.7 |
| | TransCal (ours) | **18.5** | 40.4 | 13.9 | **9.1** | 21.6 | 4.5 | **18.0** |
| | Oracle | 10.2 | 4.8 | 3.8 | 6.1 | 5.5 | 3.9 | 5.7 |

### D.2.2 Generlized to More Domain Adaptation Methods

To verify that TransCal can be generalized to recalibrate DA methods, we further conduct the experiments with the other two mainstream DA methods: DAN [13], JAN [15], and another latest DA methods: AFN [27] and BNM [5]. As shown in Figure 3, we conduct experiments on *Visda-2017* to recalibrate the above four DA methods. The results demonstrate that TransCal also performs well for these DA methods, resulting in a lower calibration error for each task.

| (a) AFN | (b) BNM | (c) DAN | (d) JAN |
|---|---|---|---|

Figure 3: ECE(%) before and after various calibration methods for more DA methods on *Visda*.

### D.2.3 Generlized to More Domain Adaptation Datasets

As shown in Figure 4, Figure 5, Figure 6 and Figure 7, TransCal also achieves much lower ECE than competitors on some domain adaptation tasks of *Office-31* and *DomainNet*.

(a) CDAN $(A \rightarrow D)$     (b) MCD $(A \rightarrow D)$     (c) DANN $(A \rightarrow D)$

(d) CDAN $(A \rightarrow W)$     (e) MCD $(A \rightarrow W)$     (f) DANN $(A \rightarrow W)$

Figure 4: ECE (%) before and after various calibration methods for several DA methods on *Office-31*.

(a) CDAN $(I \rightarrow R)$     (b) CDAN $(I \rightarrow S)$     (c) CDAN $(R \rightarrow I)$

(d) CDAN $(R \rightarrow S)$     (e) CDAN $(S \rightarrow I)$     (f) CDAN $(S \rightarrow R)$

Figure 5: ECE(%) before and after various calibration methods for CDAN on *DomainNet*.

(a) MCD ($I \rightarrow R$)  (b) MCD ($I \rightarrow S$)  (c) MCD ($R \rightarrow I$)

(d) MCD ($R \rightarrow S$)  (e) MCD ($S \rightarrow I$)  (f) MCD ($S \rightarrow R$)

Figure 6: ECE(%) before and after various calibration methods for MCD on *DomainNet*.

(a) MDD ($I \rightarrow R$)  (b) MDD ($I \rightarrow S$)  (c) MDD ($R \rightarrow I$)

(d) MDD ($R \rightarrow S$)  (e) MDD ($S \rightarrow I$)  (f) MDD ($S \rightarrow R$)

Figure 7: ECE(%) before and after various calibration methods for MDD on *DomainNet*.

### D.2.4 Evaluated by Negative Log-Likelihood (NLL)

In Section 4.2 of the main paper, we report ECE after recalibrating various domain adaptation methods on various datasets using TransCal. To verify that TransCal can also perform on other calibration metrics while only optimizing on ECE, we report the results of TransCal on NLL. As shown in Table 2, TransCal also outperforms other calibration methods when evaluated by NLL.

Table 2: NLL before and after various calibration methods for various tasks on *Office-Home*.

| Method | Transfer Task | A→C | A→P | A→R | C→A | C→P | C→R | Avg |
|---|---|---|---|---|---|---|---|---|
| | Before Cal. (Vanilla) | 3.94 | 2.13 | 2.13 | 2.97 | 2.39 | 1.87 | 2.57 |
| | IID Cal. (Temp. Scaling) | 3.13 | 1.80 | 1.71 | 2.20 | 1.75 | 1.42 | 2.00 |
| | CPCS [17] | 3.23 | 1.91 | 1.73 | 2.27 | 1.76 | 1.41 | 2.05 |
| MDD | TransCal (w/o Bias) | 2.62 | 1.62 | 1.68 | 2.31 | 1.64 | 1.45 | 1.89 |
| | TransCal (w/o Variance) | 2.51 | 1.41 | 1.37 | **2.18** | 1.54 | 1.42 | 1.74 |
| | TransCal (ours) | **2.20** | **1.31** | **1.36** | 2.20 | **1.48** | **1.40** | **1.66** |
| | Oracle | 2.13 | 1.31 | 1.35 | 1.79 | 1.47 | 1.28 | 1.56 |
| | Before Cal. (Vanilla) | 3.89 | 2.57 | 1.62 | 3.01 | 2.45 | 1.70 | 2.54 |
| | IID Cal. (Temp. Scaling) | 2.67 | 1.96 | 1.28 | 2.14 | 1.86 | 1.33 | 1.87 |
| | CPCS [17] | 2.71 | 1.97 | 1.28 | 2.09 | 1.85 | 1.33 | 1.87 |
| MCD | TransCal (w/o Bias) | 2.60 | 2.26 | 1.30 | 2.06 | 1.67 | **1.32** | 1.87 |
| | TransCal (w/o Variance) | 2.56 | **1.87** | **1.18** | 2.12 | 1.66 | 1.33 | 1.79 |
| | TransCal (ours) | **2.51** | 1.89 | 1.19 | **1.99** | **1.65** | **1.32** | **1.76** |
| | Oracle | 2.46 | 1.70 | 1.17 | 1.93 | 1.65 | 1.31 | 1.70 |

### D.2.5 Evaluated by Brier Score (BS)

Similarly, we further report the results of TransCal on BS. As shown in Table 3, TransCal outperforms its competitors on various datasets and domain adaptation methods when evaluated by BS. Note that, no matter which kind of calibration metrics we adopt to evaluate the performance, TransCal is only optimized via the proposed importance weighted expected calibration error metric.

Table 3: BS before and after various calibration methods for various tasks on *Office-Home*.

| Method | Transfer Task | A→C | A→P | A→R | C→A | C→P | C→R | Avg |
|---|---|---|---|---|---|---|---|---|
| | Before Cal. (Vanilla) | 0.780 | 0.455 | 0.455 | 0.683 | 0.542 | 0.491 | 0.568 |
| | IID Cal. (Temp. Scaling) | 0.739 | 0.442 | 0.438 | 0.630 | 0.501 | 0.452 | 0.534 |
| | CPCS [17] | 0.745 | 0.447 | 0.438 | 0.637 | 0.502 | 0.451 | 0.537 |
| MDD | TransCal (w/o Bias) | 0.699 | 0.433 | 0.436 | 0.640 | 0.491 | 0.456 | 0.526 |
| | TransCal (w/o Variance) | 0.687 | 0.422 | **0.419** | **0.628** | 0.480 | 0.452 | 0.515 |
| | TransCal (ours) | **0.647** | **0.420** | **0.419** | 0.630 | **0.473** | **0.449** | **0.506** |
| | Oracle | 0.635 | 0.419 | 0.419 | 0.577 | 0.473 | 0.432 | 0.493 |
| | Before Cal. (Vanilla) | 0.914 | 0.635 | 0.452 | 0.748 | 0.617 | 0.512 | 0.647 |
| | IID Cal. (Temp. Scaling) | 0.790 | 0.595 | 0.420 | 0.670 | 0.575 | 0.463 | 0.586 |
| | CPCS [17] | 0.796 | 0.597 | 0.421 | 0.661 | 0.573 | 0.463 | 0.585 |
| MCD | TransCal (w/o Bias) | 0.776 | 0.620 | 0.424 | 0.655 | 0.546 | **0.461** | 0.580 |
| | TransCal (w/o Variance) | 0.768 | **0.585** | **0.394** | 0.666 | 0.542 | 0.463 | 0.570 |
| | TransCal (ours) | **0.756** | 0.588 | 0.396 | **0.641** | **0.540** | 0.462 | **0.564** |
| | Oracle | 0.743 | 0.558 | 0.393 | 0.622 | 0.540 | 0.455 | 0.552 |

## D.3    More Qualitative Results.

Here, we further report more reliability diagrams for more DA tasks in Figure 8, Figure 9, Figure 10, Figure 11, Figure 12, Figure 13 respectively, showing that TransCal performs much better.

Figure 8: Reliability diagrams for the model from *Art* to *Clipart* before and after calibration.

Figure 9: Reliability diagrams for the model from *Art* to *Product* before and after calibration.

Figure 10: Reliability diagrams for the model from *Art* to *Real-World* before and after calibration.

Figure 11: Reliability diagrams for the model from *Real-World* to *Art* before and after calibration.

(a) Vanilla     (b) IID Calibration     (c) TransCal     (d) Oracle

Figure 12: Reliability diagrams for the model from *Real-World* to *Clipart* before and after calibration.

(a) Vanilla     (b) IID Calibration     (c) TransCal     (d) Oracle

Figure 13: Reliability diagrams for the model from *Real-World* to *Product* before and after calibration.

## Acknowledgements

This work was supported by the National Natural Science Foundation of China (61772299, 71690231), Beijing Nova Program (Z201100006820041), University S&T Innovation Plan by the Ministry of Education of China.