[Reviews · NeurIPS 2020]

Review 1

Summary and Contributions: In this paper, the authors proposed a new problem of cross-domain calibration. First, the authors revealed a fact that the existing domain adaptation methods improves the prediction accuracy at the cost of calibration performance. Second, the authors proposed a new method called TransCal to achieve accurate calibration under distribution shift.

Strengths: To the best of my knowledge, this is the first work tries to solve the cross domain calibration. The proposed method is well motivated, with clear theoretical motivations. The experimental analysis is comprehensive, and the proposed method is shown to be effective with different baseline DA methods and datasets.

Weaknesses: The proposed setting is very new the domain adaptation area. I appreciate that the authors try to make the calibration problem as clear as possible in such a short section, it would be better if the authors could give more detailed explanation of Section 3.1 in the Supplementary. In the equations in Section 3.2 and after, the term E_x\sim q only calculate expectation w.r.t. X, what is role of y in the equations, such as Eq (3)? From the formulation in Eq (3), it seems that the authors are making the covariate shift assumption? If so, it would better to explicitly state this assumption. I am wondering whether Eq (5) can be termed as a bias. The estimation error of the weight can be caused by random sampling if the estimator of w is unbiased. In fact, controlling the bound M of the weights may cause additional estimation bias, though the overall estimation error could be reduced.

Correctness: The proposed method should be correct under the assumptions made in the paper.

Clarity: The paper is well written.

Relation to Prior Work: Most of the relevant works are discussed in the paper. While the paper focuses on covariate shift, it would be better to cite papers in other settings such as (generalized) target/label shift to make the literature review more complete.

Reproducibility: Yes

Additional Feedback: See weakness. ------ Post Rebuttal ------ I have read the rebuttal and keep my prior rating. The authors addressed my concern and I recommend acceptance.


Review 2

Summary and Contributions: The authors address a novel issue of prediction calibration in DA which is critical for practical applications. First, the authors reveal that existing DA methods are prone to learn higher accuracy at the expense of deteriorated prediction confidence. A novel calibration metric is proposed. Further, a transferable calibration framework is proposed to obtain better target prediction calibrations with lower bias and variance. The experimental results are also comprehensive and convincing.

Strengths: 1) The paper considers the problem of target prediction calibration in domain adaptation, which is novel to the community; 2) A new Calibration Metric is proposed; 3) The quantitative and qualitative results are convincing and impressive. The bias and variance reduction strategies are shown to be effective.

Weaknesses: 1. The built-in methods, e.g., the MC-dropout, should be compared. The authors should clarify the advantage of the adopted post-hoc approaches over the buidt-in methods, e.g., [1,2]. 2. Why the proposed new Calibration Metric is reasonable? Could the authors compare to some of the typical Calibration metrics in the domain adaptation and give some discussions? 3. Could the authors provide some clarifications or comparisons or ablation study of why other bias or variance reduction strategies? For example, why use the control variate method of [22] instead of the various approaches? 4. How the source prediction calibration is affected by the proposed method? Further, the target prediction calibration of the source-only model also needs to be provided, especially in the Table 2. [1] Bayesian Uncertainty Matching for Unsupervised Domain Adaptation,2019; [2] Unsupervised Domain Adaptation via Calibrating Uncertainties,2019;

Correctness: Yes.

Clarity: Overall, the paper is well-writtten and easy to follow. But, some statements seem ambiguous, e.g., " DA models learn higher accuracy at the expense of well-calibrated probabilities".

Relation to Prior Work: Yes. The authors should provide more discussions of the built-in uncertainty calibration methods.

Reproducibility: No

Additional Feedback: 1. It would be better to caption the axis in the Fig.2 and Fig.4. 2. The results of built-in methods, e.g., the MC-dropout, should also be reported, compared, and analyzed.


Review 3

Summary and Contributions: For Unsupervised Domain Adaptation settings, the paper suggests that calibration is an important property to be transferred as well as accuracy. Moreover, referring to empirical evidence that the quantities might be at odds (that is, increase in target accuracy might result in degraded target calibration) motivates the design and analysis of post-hoc target calibration methods aimed at restoring target calibration for trained target classifiers. Lacking target labels (precluding the applicability of standard calibration methods), the paper describes such a temperature-based target calibration methods that is based on estimating a target calibration surrogate using point-wise importance-weight estimates. Methods to reduce the bias and variance of the weight estimates are incorporated in the suggested target-calibration algorithm. The proposed algorithm is evaluated extensively over several image-based data sets.

Strengths: The paper addresses an important problem and suggests a novel, practical and easy to implement (post-hoc) approach. Substantial empirical evidence is provided supporting the effectiveness of the proposed method in image-based settings.

Weaknesses: The paper introduces a general concept that may be applied in different data-settings but only presents empirical evidence of effectiveness in one setting type (image-based) - experiment on NLP data sets (sentiment analysis, named entity recognition, etc') could give more confidence regarding the general applicability of the methods, especially since importance-weight methods are prone to poor point estimates. Furthermore, missing from the experimental analysis is the effect on performance of applying the proposed target-calibration methods (especially since the trade-of was claimed).

Correctness: There seems to be an error in the derivation of the bias reduction method: below line 182 it is implicit that q(x) = 1 - p(x) but this is not necessarily so in general (with p and q being the source and target domain distributions, respectively, as stated in lines 116, 117). Most of the theoretical derivation was referred to the supplementary material.

Clarity: For the most part the paper is easy to read but it suffers occasionally in notation and clarity: First, it should explicitly state that the setting is covariate-shift (conditional label probabilities are the same across domains), otherwise the proposed methods are unfounded. Table 1 (especially the CPCS row) requires elaboration, and in several places (e.g., lines 102, 111, 170, 194) typo fixes, grammatical style, or further clarifications may be required. Finally, the lack of clarity hinders reproducibility since the resulting algorithm is not explicitly stated formally, rather through a set of equations.

Relation to Prior Work: yes.

Reproducibility: No

Additional Feedback: after going over the authors feedback I increased my score.


Review 4

Summary and Contributions: This paper considers the problem of calibrating predicted probabilities when transferring a trained model from a source domain to a target domain without any given labels. The main contribution is the TransCal approach, which is built upon temperature scaling and optimises a transferred calibration error.

Strengths: 1. The targeted problem is indeed interesting to the research community on probability calibration. 2. The proposed Transcal approach together with the bias and variance reduction methods seem to work well on confidence calibration and shows some hints for further researches.

Weaknesses: 1. From a probability calibration view, this paper focuses only on the simplest setting of confidence calibration, which is known to be less suitable for the general multi-class setting. 2. The proposed approach is only based on the temperature scaling case and leaves many closely related approaches (e.g. vector scaling/matrix scaling ) untouched in both the method and experiments. 3. A minor point is this paper also only considers covariate shifts, but, understandably, it is probably the first step to solve issues related to domain adaption.

Correctness: On the method side, the proposed Transcal approach does provide a sounding solution on top of temperature scaling. On the claims side, for reasons above, I don't think the claim of "achieving accurate calibration." is entirely achieved. A statement like "improve confidence calibration" is more suitable. Experiments suffer the same issue; the comparison is only made by confidence-ECE and only involves temperature scaling for IID calibration. It hence leaves an important question. Will a strong IID calibrator be preferable than a weak transferred calibrator? Will the results be different on another evaluation metric (whether calibration related or not, e.g. maximum ECE, accuracy, brier score)?

Clarity: This paper is mostly clear; readers with related background should understand most contents without any significant issue.

Relation to Prior Work: This paper covers related work for domain adaption and confidence calibration but doesn't include related work for multi-class probability calibration and corresponding evaluation metrics.

Reproducibility: Yes

Additional Feedback: As stated above, my main criticism for this paper is the focus is limited to confidence ECE and temperature scaling. While I appreciate the efforts on controlling the bias and variance, it would provide a more substantial impact if the methods can help other related IID calibrators. My suggestions would be hence to introduce closed related approaches like vector/matrix scaling to the framework, and further, evaluate the methods with various calibration concepts and metrics. ===========AFTER AUTHOR FEEDBACK===================== While I do appreciate the contribution by linking calibration and domain adaptation (and increased my score after hearing from other reviewers), this paper still feels limited to the problem of calibration. It would be a clear accept to me if this paper further works on true multi-class calibration and uses relevant calibration tests (such as 1910.11385). 1. As suggested in the author feedback, this paper mainly follows the definition of calibration in [16] Guo et al. 2017. While that paper did a nice introduction to the deep learning community, the proposed confidence calibration is very limited in a general multi-class setting (e.g. it merges all the classes by considering the highest probability), and particularly favours temperature scaling (e.g. call all classes by a single temperature). Therefore, as a partitioner, I am not sure if I should use the proposed approach if I want to calibrate a model when facing covariate shifts. 2. The author also stated that vector scaling and matrix scaling perform worse due to over-fitting. This is a well-known issue for both methods and can be fixed by applying suitable regularisation approaches (and outperform regularised temperature scaling for calibrating various models see 1910.12656). In fact, even for the temperature scaling, this paper has to propose a separate control method to ensure its performance. It hence feels a little unfair to state other IID scaling approaches are weaker than the proposed when no regularisation is considered. Furthermore, given the close relationship among these scaling approaches, this paper will feel more completed if the authors managed to generalise the adoption method to vector scaling and matrix scaling.

[Author Response · NeurIPS 2020]

We thank the reviewers for insightful and constructive comments. We have submitted **code** and **detailed Appdendix** .

**Common Question Q1: The covariate shift assumption.**

**A1:** Thanks for reviewers pointing out the covariate shift assumption of this paper. As a fundamental assumption of

TransCal, it is inadvertently omitted by us while writing. We will *explicitly state* it in the future version, and discuss the

*relevant papers* on covariate shift and (generalized) target/label shift to make the literature review more complete.

**Common Question Q2: Will TransCal have a lower accuracy while achieving a better calibration?**

**A2:** As a post-hoc method that softens the overconfident probabilities but *keeps the probability order over classes*,

TransCal maintains the **same accuracy** with that before calibration, while achieving a lower ECE (Fig. 1(b)).

**R1.1: Whether Eq. (5) can be termed as a bias?**

Realizing the gap between the importance weights estimated by LogReg [38, 1, 5] and the (*unknown*) ground-truth

ones, we proposed to control the bound $M$ of the weights to reduce the overall estimation error. Further, as reported in

Line 235, we ran each experiment 10 times with different sampling data to mitigate the problem of random sampling.

**R2.1: The advantage of the adopted post-hoc approaches over the built-in methods, *e.g.* MC-dropout.**

TransCal maintains the **same accuracy** with that before calibration while built-in methods (*e.g.* MC-dropout) may

*degrade* prediction accuracy (Fig. 1(b)), and they have to modify the network architecture (*e.g.* adding dropout layers).

| Calibration Method | A→C | A→P | A→R | C→A | C→P | C→R | Avg |
|---|---|---|---|---|---|---|---|
| Before Cal. (Vanilla) | 40.2 | 26.4 | 17.8 | 35.8 | 23.5 | 21.9 | 27.6 |
| IID Cal. (MC-dropout) | 33.1 | 21.3 | 15.0 | 24.2 | 20.5 | 13.2 | 18.8 |
| IID Cal. (Matrix Scaling) | 44.7 | 28.8 | 19.7 | 36.1 | 25.4 | 24.1 | 29.8 |
| IID Cal. (Vector Scaling) | 34.7 | 18.0 | 11.3 | 23.4 | 15.4 | 11.5 | 19.4 |
| IID Cal. (Temp. Scaling) | 28.3 | 17.6 | 10.1 | 21.2 | 13.2 | 8.2 | 16.4 |
| **TransCal (ours)** | 13.2 | 9.9 | 5.2 | 21.2 | 8.1 | 6.4 | 10.7 |

(a) ECE (%) on *Office-Home* for DA method CDAN     (b) ECE vs. Accuracy     (c) *Multi-Domain Sentiment*

**R2.2: Why the proposed new Calibration Metric is reasonable?**

Among the three typical calibration metrics, BS *conflates* accuracy with calibration and NLL may *over-emphasize* tail

probabilities [31], thus we proposed TransCal based on the intuitive and informative one: ECE (Paragraph at Line 149).

**R2.3: Why we use the control variate method of [22] instead of the various approaches?**

As a *non-intrusive* and *parameter-free* method, control variate is the mainstream, simple and effective variance reduction

method. Besides, we further developed *serial* control variate method backed by a theoretical analysis in B.2 of *Appendix*.

**R2.4: How will TransCal perform on the source prediction? The calibration result of the source-only model.**

TransCal performs well on source prediction and source-only model (ECE decreases $\sim 20\%$ than that before calibration).

**R3.1: Experiments on NLP datasets.**

TransCal performs well in 12 transfer tasks of a popular NLP dataset: *Amazon Multi-Domain Sentiment* (Fig. 1(c)).

**R3.2: The missing experimental analysis on performance of applying the proposed target calibration method.**

See common question Q2. We believe there is no need to report the **same accuracy** before and after calibration.

**R3.3: There seems to be an error in the derivation of the bias reduction method.**

We use LogReg to estimate density ratio from a logistic regression classifier that separates examples from the source and

target domains as in Eq. (4). We clarify that $q(x) = 1 - p(x)$ below Line 182 is the output of LogReg, indicating the

probability of the target domain that $x$ belongs to. Notations in Line 182 will be updated to avoid such *misunderstanding*.

**R3.4: Minor issues on related works (CPCS elaboration), typos, grammar and formally stated algorithm.**

Thanks for the valuable suggestions from the reviewer. We will *fully* address these minor issues in the future version.

**R4.1: This paper focuses only on the simplest setting of confidence calibration.**

As the first transferable calibration work for Domain Adaptation (DA), we adopt the fundamental and mainstream

confidence calibration. Thanks for your valuable suggestion, pointing out our future work on more complex settings.

**R4.2: The results of calibration methods with vector scaling/matrix scaling.**

Both Vector Scaling and Matrix Scaling underperform TransCal and Temp Scaling (Table 1(a)). Matrix Scaling works

even worse than the Vanilla model due to overfitting, which was also observed in the results of Guo *et al.*, [16] (Table 2).

**R4.3: Will a strong IID calibrator be preferable than a weak transferred calibrator?**

Besides the result of IID calibrator Temp Scaling given in Table 2, we add the results of competitive IID calibrators, *e.g.*

Vector Scaling, Matrix Scaling and MC-dropout (Table 1(a)). They all underperform TransCal in the Non-IID setup.

**R4.4: Will the results be different on another evaluation metric, *e.g.* maximum ECE, accuracy, Brier Score?**

See common question Q2 about evaluating on accuracy. The results on Brier Score, NLL and Reliability Diagrams

were already given in D.2.5, D.2.4 and D.3 of *Appendix*. They *consistently* demonstrate the efficacy of TransCal.

[Meta-Review · NeurIPS 2020]

Reviewers agree the paper addressed an important new problem on cross-domain calibration. The motivation is strong, the proposed method is easy to implement and shown to be effective (in the setting of the paper). The authors are highly encouraged to take into consideration reviewers' concerns on the experiments and limitation of the settings, in the revision of the paper.